# Optimal Fair Learning Robust to Adversarial Distribution Shift

## Abstract

Previous work in fair machine learning has characterised the Fair Bayes Optimal Classifier (BOC) on a given distribution for both deterministic and randomized classifiers. We study the robustness of the Fair BOC to adversarial noise in the data distribution. Kearns and Li [1988] implies that the accuracy of the deterministic BOC without any fairness constraints is robust (Lipschitz) to malicious noise in the data distribution. We demonstrate that their robustness guarantee breaks down when we add fairness constraints. Hence, we consider the randomized Fair BOC, and our central result is that its accuracy is robust to malicious noise in the data distribution. Our robustness result applies to various fairness constraints— Demographic Parity, Equal Opportunity, Predictive Equality. Beyond robustness, we demonstrate that randomization leads to better accuracy and efficiency. However, we show that the randomized Fair BOC is nearly-deterministic, and gives randomized predictions on at most one data point, hence availing numerous benefits of randomness, while using very little of it.

## 1 Introduction

The effectiveness of machine learning models has resulted in improved efficiency across multiple domains but has also raised concerns about their fairness and possible amplification of biases in their training data [Barocas et al., 2019]. When machine learning models are used to make decisions that skew the distribution of important economic resources or reinforce stereotypes, they compound disparities to cause social and economic harm. Fair classification has been an important topic of research, and binary fair classification where the model makes yes/no decisions algorithmically is a simple yet challenging setting to study foundational questions in optimal fair classification [Menon and Williamson, 2018b]. In group-fair classification, each data point has certain sensitive attributes indicating the demographic group(s) to which it belongs (e.g., race, gender). Popular notions of group-fairness such as statistical or demographic parity, equal opportunity, equalized odds, and predictive parity are all motivated by the binary fair classification setting. Demographic parity prescribes the positivity rates to be equal across different groups (e.g., race, gender), whereas equal opportunity prescribes the true positive rates to be equal across different groups [Dwork et al., 2012, Hardt et al., 2016]. Previous work has looked at various trade-offs between accuracy and fairness as well as the difficulty in satisfying multiple fairness constraints simultaneously [Celis et al., 2020]. Previous work has also mathematically characterized the Fair Bayes Optimal Classifier (BOC), namely, the optimal deterministic classifiers for maximizing accuracy subject to group-fairness constraints based such as demographic parity and equal opportunity [Menon and Williamson, 2018a, Chzhen et al., 2019, Celis et al., 2021, Zeng et al., 2022]. Pre-processing or re-weighing for training data imbalances, in-processing by fairness-constrained training loss, and post-processing a model's predictions for balanced outcomes are three known ways to realize fair and accurate classifiers in practice [Kamiran and Calders, 2012, Agarwal et al., 2018, Barocas et al., 2019].

Biased or corrupted training data is a primary cause of unfairness in model predictions or outcomes. Moreover, robustness of a machine learning model under bias or corruption in the data distribution has been a more pragmatic concern that predates the research on fair machine learning. Learning robust classifiers is important because training and test distributions are not always identical and the training data may contain noise and malicious corruptions during data collection, curation, and annotation. Robustness of fair classifiers under bias/shift in the data distribution is a well studied issue in fair machine learning literature. Akpinar et al. [2022] empirically study the robustness of BOC and Fair BOC on synthetic data distributions and provide a sandbox tool for stress-testing fair classifiers. Sharma et al. [2023] and Ghosh et al. empirically study robustness of fair classifiers under data bias on semi-synthetic real-world datasets (i.e., real-world datasets with synthetically injected bias/shift). In both these papers, Exponentiated Gradient Reduction (EGR) or ExpGrad [Agarwal et al., 2018] stands out for its better robustness under data bias/shift, and it is inherently a randomized classifier.

A particularly compelling and illustrative practical example for fair binary classification with maliciously corrupted training data is that of hate speech classifiers. Hate speech classifiers are known to exhibit biases against the same vulnerable demographics they were supposed to protect in online forums. For example, text in African American English (AAE) has higher likelihood of being misreported as hate speech and even proper mentions of group identifiers such as 'gay' or 'black' get misreported as toxic or prejudiced. Moreover, the training data taken from online forums that is used to train hate speech classifiers contains societal biases of novice human annotators as well as malicious attempts made to bypass existing classifiers or filters used in data collections and annotation process [Davani et al., 2023, Davidson, 2023]. Maliciously corrupted training data makes it difficult to train fair hate speech classifiers with robust accuracy and fairness guarantees that would be retained after real-world deployment [Davani et al., 2023, Davidson, 2023, Hartvigsen et al., 2022, Harris et al., 2022].

Classification under malicious noise is a theoretically challenging direction on its own, even without any fairness constraints. Balcan and Haghtalab [2020] survey research directions that originate from the work of Kearns and Li [1988], but focus on the hardness of learning linear classifiers under malicious noise and recent results that get around it. Unlike previous works on learning from malicious noise that consider any hypothesis class or a specific one such as linear classifiers, we consider the hypothesis class of all binary classifiers, deterministic as well as randomized. Although previous work in fair machine learning has extensively studied the Fair BOC and fair pre-/in-/post-processing methods to achieve best possible fairness-accuracy trade-offs, their fairness and accuracy guarantees may not hold when training data is biased or contaminated and does not match test data. Adversarial or unknown bias in data makes it important to study the robustness of fairness and accuracy guarantees of the Fair BOC.

The seminal work of Kearns and Li [1988] shows the robustness (of accuracy) to malicious noise of any deterministic hypothesis class (without fairness constraints) in terms of a Lipschitz condition, i.e., given two similar distributions, the accuracy of the optimal classifier on each distribution is also similar. In particular, their robustness guarantee also carries over to the deterministic BOC. In contrast, more recent findings by Konstantinov and Lampert [2022] reveal a concerning vulnerability: incorporating fairness constraints can render certain deterministic hypothesis classes non-robust to adversarial noise. This gap in understanding necessitates an investigation into the robustness of Fair BOC's under adversarial distribution shift, which in turn is the focus of this paper.

## 1.1 Overview of Our Results

We summarize our key contributions.

- We demonstrate in Claim 1 (Section 3.1) that the deterministic Fair BOC is not robust to adversarial noise, corroborating Konstantinov and Lampert [2022].

Our main results prove the robustness of randomized Fair BOC's.

- We prove in Theorems 1 (Section 3.2), 2 and 3 (Section 4) that the accuracy of the randomized Fair BOC is robust to malicious noise across three popular fairness notions (Demographic Parity, Equal Opportunity, and Predictive Equality). This robustness is charac-

terized by a (local) Lipschitz property, where the Lipschitz constant depends on the distribution [Yang et al., 2020].

- Toward this end, we first prove in Claims 2, 3, and 7 (Sections 3.2 and 4) that a fixed hypothesis maintains comparable accuracy and fairness across two similar distributions. This, however, does not imply our main results since the Fair BOC may change significantly for neighboring distributions. We establish the Lipschitz property using a more sophisticated analysis of the specific structure of the randomized Fair BOC.

In addition to robustness, randomization confers multiple advantages.

- Claim 1 demonstrates that the Randomized Fair BOC can outperform its deterministic counterpart in accuracy by $0.5 - \epsilon$ (for any $\epsilon > 0$). We complement this with a tightness result in Claim 6 (Appendix B).

- The Randomized Fair BOC can be computed in polynomial time, whereas we prove in Claim 5 (Appendix B) that computing the deterministic Fair BOC is NP-complete.

Randomization is a very natural and useful resource for fairness as ties are often broken by a random coin toss. However, when it brings arbitrariness to critical decisions, it needs to be used judiciously and sparingly [Creel and Hellman, 2021, Rosenblatt and Witter, 2024, Cooper et al., 2024]. A key property of the randomized Fair BOC is that it is *nearly deterministic*, being randomized at most on a single point in the domain and deterministic elsewhere. Thus, in a sense, we have the best of both worlds, preserving the benefits of randomization, while using very little of it.

We present the problem formulation in Section 2. More detailed comparison with most relevant previous work is given in Appendix A, and we conclude in Section 5.

## 2 Problem Formulation

We are given a discrete distribution $\mathcal{P}$ over $\mathcal{X} \times \mathcal{Z} \times \mathcal{Y}$, where $\mathcal{Z} = \{A, D\}$ represents the protected group membership ($A$ denotes the advantaged group, and $D$ denotes the disadvantaged group)[1], $\mathcal{X}$ represents all the other features, and $\mathcal{Y} = \{0, 1\}$ represents the binary label set (we adopt the standard convention of associating the label 1 with success or acceptance). A randomized classification rule $f$ is a function $f : \mathcal{X} \times \mathcal{Z} \to [0, 1]$, where $f(x, z)$ denotes the probability of a feature vector or instance $(x, z) \in \mathcal{X} \times \mathcal{Z}$ being mapped to 1. A deterministic classifier is defined similarly, however the output of $f(x, z)$ is restricted to $\{0, 1\}$. We consider the standard 0-1 loss function $\ell_{0-1}$[2], whose expected value is given by $\mathcal{L}(f, \mathcal{P}) = \mathbb{E}[\ell_{0-1}(f)] = \Pr[f(X, Z) \neq Y]$, where the probability is over $(X, Z, Y) \sim \mathcal{P}$[3]. As is standard, we define accuracy as $\mathrm{Acc}(f, \mathcal{P}) = 1 - \mathcal{L}(f, \mathcal{P})$.

In a fairness-aware learning problem, we want to find an accurate classifier on a given distribution that also satisfies some fairness constraints. Our work considers 3 of the most popular notions of fairness (Demographic Parity, Equal Opportunity, Predictive Equality). We present our proofs for Demographic Parity in the main body, and defer the proofs of the other 2 notions to Appendix 4. We state the Demographic Parity definition below [Dwork et al., 2012].

**Definition 1** (Demographic Parity). Denote the selection rate for group $z$ by $\mathrm{r}_z(f, \mathcal{P}) = \Pr[f(X, Z) = 1 \mid Z = z]$. $f$ satisfies Demographic Parity[4] if the selection rates are equal across both groups, i.e., $\mathrm{r}_A(f, \mathcal{P}) = \mathrm{r}_D(f, \mathcal{P})$. We quantify the unfairness of $f$ as the difference in selection rates across groups , i.e., $\mathrm{Unf}_{\mathrm{DP}}(f, \mathcal{P}) = |\mathrm{r}_A(f, \mathcal{P}) - \mathrm{r}_D(f, \mathcal{P})|$.

---

[1]Our results also hold when there are multiple groups, but for ease of exposition, we restrict our analysis to the case of 2 groups.

[2]Using the same proof techniques, our results also hold for the more general loss function $\ell_\alpha$, known in literature as cost-sensitive risk [Menon and Williamson, 2018b], that assigns a weight $\alpha$ to False Positive errors, and a weight $(1 - \alpha)$ to False Negative errors. However, for simplicity, we restrict our analysis here to $\ell_{0-1}$.

[3]Henceforth, all probabilities will be over $(X, Z, Y) \sim \mathcal{P}$, unless explicitly stated.

[4]Classifiers satisfying DP will be often be referred to as DP-fair.

## 2.1 Fair Bayes Optimal Classifier

Given a distribution $\mathcal{P}$, the optimal (accuracy-maximizing) classifier $f^*$ (the BOC) is given by $f^*(x, z) = \mathcal{T}_{\frac{1}{2}}(\Pr[Y = 1 \mid X = x, Z = z])$, where $\mathcal{T}_\gamma(\beta)$ is the threshold function that outputs 1 if $\beta \geq \gamma$, and 0 otherwise. We call the term $\beta$ in the expression above the score or success probability of a point $(x, z)$, and formally define it below.

**Definition 2** (Score). The score $\mathcal{S}$ of a point $(x, z)$ is the probability that it has label 1, i.e., $\mathcal{S}(x, z) = \Pr[Y = 1 \mid (X = x, Z = z)]$.

The BOC basically accepts a point if its score is $\geq \frac{1}{2}$, and rejects it otherwise. Note that the BOC as described above is deterministic, and allowing for randomized classifiers will not provide any increase in accuracy. However, when fairness constraints are involved, the picture is more complicated, and it turns out that allowing for randomization actually can lead to a big jump in accuracy. To see how randomized Fair BOC's can improve the accuracy of their deterministic counterparts, let us look at an example from Agarwal and Deshpande [2022].

**Example 1** (Accuracy jump in Randomized Fair BOC's). Consider the following distribution $\mathcal{P}^5$ over $\mathcal{X} \times \mathcal{Z} \times \mathcal{Y}$, where $\mathcal{X} = \{x_1, x_2\}$ ($\mathcal{P}, \mathcal{S}(x, z) = (p, q)$ denotes that $\mathcal{P}(x, z) = p$, and $\mathcal{S}(x, z) = q$).

$$\mathcal{P}, \mathcal{S}(x_1, A) = (0.5, 0.75) \qquad \mathcal{P}, \mathcal{S}(x_1, D) = (0.25, 0.5)$$
$$\mathcal{P}, \mathcal{S}(x_2, A) = (0, 0) \qquad \mathcal{P}, \mathcal{S}(x_2, D) = (0.25, 0)$$

There are only 2 deterministic classifiers satisfying DP, either the constant 1 classifier $f_1$, or the constant 0 classifier $f_0$, with $\mathcal{L}(f_1) = \mathcal{L}(f_0) = \frac{1}{2}$. On the other hand, consider the following randomized classifier $f$, where $f(x_1, A) = \frac{1}{2}, f(x_1, D) = 1, f(x_2, A) = f(x_2, D) = 0$. It is easy to see that $f$ satisfies DP, and $\mathcal{L}(f) = \frac{3}{8}$, hence improving over the accuracy of the deterministic DP-fair BOC's $f_0$ and $f_1$.

Given a distribution $\mathcal{P}$, Agarwal and Deshpande [2022] characterize the DP-Fair BOC (the optimal classifier subject to DP constraints) on a given distribution, which we now describe. We first present some of their terminology.

**Definition 3** (Cell). Consider a randomized partition of the feature space $\mathcal{X} \times \mathcal{Z}$ into multiple disjoint components. We call these components cells, and denote a cell by $\mathcal{C}$.

One can also define the score of a cell, in the same way as we had defind the score of a point. We have already seen the BOC that thresholds based on scores. Randomized classifiers give us the ability to threshold by probability mass, instead of just thresholding by scores. To explain this better, we introduce the notion of group-wise sorted cells.

**Definition 4** (Group-wise Sorted Cells). Define $\mathcal{C}_z = \bigcup_{x \in \mathcal{X}} \mathcal{C}_{x,z}$, where the component cells of $\mathcal{C}_A$ and $\mathcal{C}_D$ are arranged in descending order of scores $\mathcal{S}$. If two or more cells from the same group have the same score, any ordering within them is acceptable.

By $\mathcal{C}_z(t)$, denote the topmost cells of $\mathcal{C}_z$ comprising of $t$ fraction of the total probability mass of $\mathcal{C}_z$. Note that this may involve splitting a cell into 2 parts randomly. For example, in Example 1, $\mathcal{C}_A(\frac{1}{2})$ would involve splitting $\mathcal{C}_{x_1,A}$ into two equal parts randomly. However, in the deterministic setting, only $\mathcal{C}_A(0)$ and $\mathcal{C}_A(1)$ are defined, and not $\mathcal{C}_A(\frac{1}{2})$. By $\tilde{\mathcal{T}}_t$, we denote the mass threshold classifier that accepts exactly $\mathcal{C}_z(t)$ for $z \in \mathcal{Z}$. In Example 1, the randomized classifier $f$ is the mass-threshold classifier $\tilde{\mathcal{T}}_{\frac{1}{2}}$.

**Definition 5** (Score Boundaries). Consider the component cells of groupwise sorted $\mathcal{C}_A$ and $\mathcal{C}_D$. Then, the score boundaries denote the set $\mathcal{I} = \mathcal{I}_A \cup \mathcal{I}_D$, where $\mathcal{I}_z$ consists of all the boundary points between component cells in $\mathcal{C}_z$.

**Definition 6** (Merged Cells). Consider any $r_i \in \mathcal{I}$ in sorted order, and define a merged cell $\mathcal{C}_i$ as $\mathcal{C}_i = \mathcal{A}(\tilde{\mathcal{T}}_{r_i}) - \mathcal{A}(\tilde{\mathcal{T}}_{r_{i-}})$, where $\mathcal{A}(f)$ denotes the instances accepted by $f$, and $r_{i-}$ denotes the element in $\mathcal{I}$ preceding $r_i$.

---

[5]Note that specifying a distribution over $\mathcal{X} \times \mathcal{Z} \times \mathcal{Y}$ is equivalent to specifying a distribution over $\mathcal{X} \times \mathcal{Z}$ along with the scores for every instance $(x, z) \in \mathcal{X} \times \mathcal{Z}$.

**Characterization** Given a distribution $\mathcal{P}$ over $\mathcal{X} \times \mathcal{Z} \times \mathcal{Y}$, the DP-Fair BOC $f_{\mathcal{P}}^{\mathrm{DP}}$ is given by the mass-threshold classifier $\tilde{\mathcal{T}}_{r'}$, where $r' = r_i \in \mathcal{I}$ is the unique $i$ such that $\mathcal{S}(C_i) \geq 0.5$, and $\mathcal{S}(C_{i+}) < 0.5$, where $r_{i+}$ denotes the element in $\mathcal{I}$ after $r_i$. Note that the DP-Fair BOC needs to use randomization on at most one cell in the whole domain, since the candidate $r'$ values lie in $\mathcal{I}$. Hence, to evaluate the Fair BOC, instead of considering the hypothesis class of all randomized classifiers, it is sufficient to consider the hypothesis class of classifiers that are randomized on at most one element in the domain.

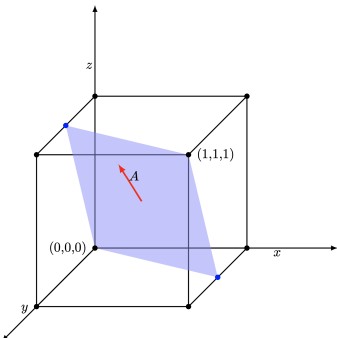

Figure 1: If the feature space $\mathcal{X} \times \mathcal{Z}$ has cardinality $n$, then the hypothesis class of all randomized classifiers $\mathcal{H}$ is the hypercube $[0,1]^n$. Similarly, the hypothesis class of all deterministic classifiers is $\{0,1\}^n$. A fairness criterion is a linear constraint (this may not be true of all fairness criteria, but is true of the well-known ones that we study in this paper), which can be represented by a hyperplane $\mathcal{F}$. Also, accuracy $A$ is a linear objective, implying that the Fair BOC is the point in $\mathcal{H} \cap \mathcal{F}$ maximizing $A$. We illustrate this in 3-dimensions here.

## 3 Robustness to Adversarial Distribution Shift

We study the robustness of the DP-Fair BOC to adversarial distribution shift. We show that given 2 similar distributions $\mathcal{P}, \mathcal{P}'$ (similarity measured by TV distance), the accuracy of the DP-Fair BOC on the respective distributions is similar (satisfies local Lipschitzness). Note that DP-Fair BOC in the deterministic case does not exhibit such a robustness property, as we demonstrate in the following example.

### 3.1 Non-Robustness of the Deterministic Fair BOC

**Claim 1** (Non-Robustness of Deterministic Fair BOC's). *Given $\epsilon > 0$, there exist $\mathcal{P}, \mathcal{P}'$ with $TV(\mathcal{P}, \mathcal{P}') \leq \epsilon$, such that the deterministic DP-Fair BOC's $f, f'$ on $\mathcal{P}, \mathcal{P}'$, respectively, satisfy $|Acc(f, \mathcal{P}) - Acc(f', \mathcal{P}')| \geq \Omega(1)$.*

*Proof.* Consider the following distribution $\mathcal{P}$, with $\mathcal{X} = \{x_1, x_2\}$.

$$\mathcal{P}, \mathcal{S}(x_1, A) = (0.25, 1) \qquad\qquad \mathcal{P}, \mathcal{S}(x_1, D) = (0.25, 1)$$
$$\mathcal{P}, \mathcal{S}(x_2, A) = (0.25, 0) \qquad\qquad \mathcal{P}, \mathcal{S}(x_2, D) = (0.25, 0)$$

Consider the (deterministic) classifier $f$, with $f(x_1, A) = f(x_1, D) = 1, f(x_2, A) = f(x_2, D) = 0$. It is easy to see that $f$ satisfies DP, and $\mathrm{Acc}(f) = 1$, implying that $f$ is the DP-Fair BOC in both the deterministic and randomized settings. Consider the neighboring distribution $\mathcal{P}'$ as follows, for small $\epsilon$.

$$\mathcal{P}', \mathcal{S}(x_1, A) = (0.25, 1) \qquad\qquad \mathcal{P}', \mathcal{S}(x_1, D) = (0.25 + \epsilon, 1)$$
$$\mathcal{P}', \mathcal{S}(x_2, A) = (0.25, 0) \qquad\qquad \mathcal{P}', \mathcal{S}(x_2, D) = (0.25 - \epsilon, 0)$$

There are only 2 deterministic classifiers satisfying DP, either the constant 1 classifier $f_1$, or the constant 0 classifier $f_0$, with $\mathcal{L}(f_1) = \frac{1}{2} + \epsilon$, and $\mathcal{L}(f_0) = \frac{1}{2} - \epsilon$, implying that $f_1$ is the DP-Fair BOC in the deterministic setting. Hence, the difference in accuracy of the deterministic DP-Fair BOC on arbitrarily close $\mathcal{P}, \mathcal{P}'$ is almost 0.5, demonstrating the non-robustness of deterministic classifiers to distribution shift. $\qquad\square$

## 3.2 Robustness of the Randomized Fair BOC

Now we state our main result.

**Theorem 1** (Robustness of DP-Fair BOC). *Given distributions* $\mathcal{P}, \mathcal{P}'$ *with* $TV(\mathcal{P}, \mathcal{P}') = \epsilon$, *we have*

$$\left| Acc(f_{\mathcal{P}}^{DP}, \mathcal{P}) - Acc(f_{\mathcal{P}'}^{DP}, \mathcal{P}') \right| \leq \epsilon \left( 1 + \frac{\max\left(\mathcal{P}(A), \mathcal{P}'(A)\right)}{\min(\mathcal{P}(A), \mathcal{P}'(A))} + \frac{\max\left(\mathcal{P}(D), \mathcal{P}'(D)\right)}{\min(\mathcal{P}(D), \mathcal{P}'(D))} \right).$$

*Remark.* Note that the Lipschitz constant will blow up if the masses of either group becomes very small. Similar terms in the denominator will naturally feature in all our bounds. As such, robustness is not satisfied at such extremal points.

We first state Lemmas 1 and 2, and Claim 2 that will help us prove Theorem 1. We defer their proofs to Appendix B. Lemma 1 shows that one can decompose a transition from distribution $\mathcal{P}$ to distribution $\mathcal{P}'$ with distance $\epsilon$ into a sequence of elementary transitions from $\mathcal{P}_{i-1}$ to $\mathcal{P}_i$ with distance $\epsilon_i$ such that $\epsilon = \sum_i \epsilon_i$ and for every $i$, the only difference between $\mathcal{P}_{i-1}$ and $\mathcal{P}_i$ is that mass is transferred from exactly one element of the domain to another.

**Lemma 1** (Decomposition into Elementary Transitions). *Given distributions* $\mathcal{P}, \mathcal{P}'$ *with* $TV(\mathcal{P}, \mathcal{P}') = \epsilon$, *there exist distributions* $\mathcal{P}_0, \mathcal{P}_1, \ldots, \mathcal{P}_n$ *(for some* $n$, *with* $\mathcal{P} = \mathcal{P}_0$, $\mathcal{P}' = \mathcal{P}_n$*), such that the following two conditions hold:*

1. *Decomposability:* $TV(\mathcal{P}_{i-1}, \mathcal{P}_i) = \epsilon_i$, $\sum_{i=1}^{n} \epsilon_i = \epsilon$, *and in the transition* $\mathcal{P}_{i-1} \to \mathcal{P}_i$, $\epsilon_i$ *mass moves from some instance* $a_i$ *to some* $b_i$ *(*$a_i, b_i \in \mathcal{X} \times \mathcal{Z}$, *all other elements remain constant).*

2. *Monotonicity:* *If* $\mathcal{P}(A) \leq \mathcal{P}'(A)$, *then for every* $1 \leq i < n$, $\mathcal{P}_i(A) \leq \mathcal{P}_{i+1}(A)$ *and* $\mathcal{P}_i(D) \geq \mathcal{P}_{i+1}(D)$; *otherwise,* $\mathcal{P}_i(A) \geq \mathcal{P}_{i+1}(A)$ *and* $\mathcal{P}_i(D) \leq \mathcal{P}_{i+1}(D)$.

Claim 2 roughly states that given 2 similar distributions $\mathcal{P}, \mathcal{P}'$, the accuracy and DP-unfairness of any fixed hypothesis is similar on both $\mathcal{P}, \mathcal{P}'$. Such a property is useful when we want a guarantee that if we train a classifier on the corrupted distribution $\mathcal{P}'$, the performance of the classifier on the actual distribution $\mathcal{P}$ will be similar to that on $\mathcal{P}'$.

**Claim 2** (Accuracy, DP Shift for Fixed Hypothesis). *Given distributions* $\mathcal{P}, \mathcal{P}'$, *such that* $TV(\mathcal{P}, \mathcal{P}') \leq \epsilon$, *any hypothesis* $f$ *satisfies the following two properties:*

1. $|Acc(f, \mathcal{P}) - Acc(f, \mathcal{P}')| \leq \epsilon$.

2. $|Unf_{DP}(f, \mathcal{P}) - Unf_{DP}(f, \mathcal{P}')| \leq \epsilon \left( \frac{1}{\min(\mathcal{P}(A), \mathcal{P}'(A))} + \frac{1}{\min(\mathcal{P}(D), \mathcal{P}'(D))} \right)$

We also use Lemma 2 for our main result .

**Lemma 2.** *Given any* $\mathcal{P}, f$, *and* $\mathcal{P}', f'$ *such that* $TV(\mathcal{P}, \mathcal{P}') = \epsilon$, *if* $f'(q)$ *differs from* $f(q)$ *by* $\Delta f(q)$ *(and is identical elsewhere), then*

$$|Acc(f, \mathcal{P}) - Acc(f', \mathcal{P}')| \leq |\mathcal{P}(q)(2\mathcal{S}(q) - 1)\Delta f(q)| + \epsilon.$$

Now move on to the proof of our main theorem.

*Proof of Theorem 1.* Armed with these lemmas, we first establish the claim of the theorem for the special case where the transition from $\mathcal{P}$ to $\mathcal{P}'$ is elementary in that the only difference between the two distributions is that there are two elements $a$ and $b$ that have $\epsilon$ more mass and $\epsilon$ less mass, respectively, in $\mathcal{P}$ as compared to $\mathcal{P}'$ (all other elements have the same mass in the two distributions). At the end, we invoke Lemma 1 and transitivity to establish the general theorem statement.

Consider the transfer of $\epsilon$ mass from $a$ to $b$ in a continuous manner. During this process, either the cell corresponding to element $a$ will monotonically increase in score or monotonically decrease in score[6]. The same holds for the cell corresponding to element $b$. The scores of all other cells will remain the same. In the following argument, we assume that the score of the cell of $a$ decreases

---

[6]In case the cell corresponding to $a$ has score of 0 or 1, it's score will remain unchanged, and this case is trivially covered by our argument.

242  monotonically and that of $b$ increases monotonically. All of the arguments are analogous for the
243  remaining three cases.

244  We break down the $\epsilon$ mass transfer into smaller increments. At any point, let $\tilde{\mathcal{P}}$ be the distribution at
245  the start of this increment (so, $\tilde{\mathcal{P}} = \mathcal{P}$ initially) and $\tilde{\mathcal{P}}'$ be the distribution at the end of this increment
246  (so, $\tilde{\mathcal{P}}' = \mathcal{P}'$ finally). For an incremental mass transfer, we analyze how the DP BOC changes from
247  $f_{\tilde{\mathcal{P}}}^{\text{DP}}$ to $f_{\tilde{\mathcal{P}}'}^{\text{DP}}$. Since the mass transfer is from element $a$ to $b$, it follows that both $\tilde{\mathcal{P}}(A)$ and $\tilde{\mathcal{P}}'(A)$ lie
248  between $\mathcal{P}(A)$ and $\mathcal{P}'(A)$ while both $\tilde{\mathcal{P}}(D)$ and $\tilde{\mathcal{P}}'(D)$ lie between $\mathcal{P}(D)$ and $\mathcal{P}'(D)$. We consider
249  the largest mass transfer $\delta\epsilon$ until one of the two following events occur.

250      1. Equal-score event: The cell of $a$ has the same score as the adjacent cell lower in the sorted
251         order or the cell of $b$ has the same score as the adjacent cell higher in the sorted order.

252      2. Threshold event: The score of a merged cell containing $a$ or $b$ becomes exactly 0.5.

253  **Bounding the accuracy change for $\delta\epsilon$:**  Note that by the choice of $\delta\epsilon$, during the transfer $\delta\epsilon$, all
254  the cells remain in the same order in both groups; furthermore, all masses and scores of all cells
255  other than the ones containing $a$ or $b$ remain the same during the transfer. By part 2 of Claim 2,

$$\delta\text{Unf}_{\text{DP}} = \left| \text{Unf}_{\text{DP}}(f_{\tilde{\mathcal{P}}}^{\text{DP}}, \tilde{\mathcal{P}}) - \text{Unf}_{\text{DP}}(f_{\tilde{\mathcal{P}}}^{\text{DP}}, \tilde{\mathcal{P}}') \right|$$

$$\leq \delta\epsilon \left( \frac{1}{\min(\tilde{\mathcal{P}}(A), \tilde{\mathcal{P}}'(A))} + \frac{1}{\min(\tilde{\mathcal{P}}(D), \tilde{\mathcal{P}}'(D))} \right)$$

$$\leq \delta\epsilon \left( \frac{1}{\min(\mathcal{P}(A), \mathcal{P}'(A))} + \frac{1}{\min(\mathcal{P}(D), \mathcal{P}'(D))} \right)$$

256  Since $\text{Unf}_{\text{DP}}(f_{\tilde{\mathcal{P}}}^{\text{DP}}, \tilde{\mathcal{P}}) = 0$, we know that $\delta\text{Unf}_{\text{DP}} = \text{Unf}_{\text{DP}}(f_{\tilde{\mathcal{P}}}^{\text{DP}}, \tilde{\mathcal{P}}') = \left| \text{r}_A(f_{\tilde{\mathcal{P}}}^{\text{DP}}, \tilde{\mathcal{P}}) - \text{r}_D(f_{\tilde{\mathcal{P}}}^{\text{DP}}, \tilde{\mathcal{P}}') \right|$.
257  Consider the cell $q$ that is split in the middle by the threshold corresponding to $f_{\tilde{\mathcal{P}}}^{\text{DP}}$ (for now, assume
258  $q \in D$). Since neither the equal-score event nor the 0.5-score event occur, we see that after the
259  transition, the boundary of $f_{\tilde{\mathcal{P}}}^{\text{DP}}$ intersecting $q$ is $\delta\text{Unf}_{\text{DP}}$ away from the boundary in group $A$. To
260  modify $f_{\tilde{\mathcal{P}}}^{\text{DP}} \to f_{\tilde{\mathcal{P}}'}^{\text{DP}}$, we therefore need to move to move the boundary at $q$ by $\delta\text{Unf}_{\text{DP}}$ so that the
261  boundaries in both groups align and DP is satisfied (the classifier remains the same apart from its
262  action on $q$). The change in function value on element $q$, which we denote by $|\Delta f(q)|$, is bounded
263  by $\delta\text{Unf}_{\text{DP}} \frac{\tilde{\mathcal{P}}(D)}{\tilde{\mathcal{P}}(q)}$, after scaling (since $\tilde{\mathcal{P}}(D)\delta\text{Unf}_{\text{DP}} = |\Delta f(q)|\tilde{\mathcal{P}}(q)$). At the end of the $\delta\epsilon$ mass
264  transfer, by Lemma 2, the change in accuracy of the optimal fair classifier is given by

$$\left| \text{Acc}(f_{\tilde{\mathcal{P}}}^{\text{DP}}, \tilde{\mathcal{P}}) - \text{Acc}(f_{\tilde{\mathcal{P}}'}^{\text{DP}}, \tilde{\mathcal{P}}') \right| \leq \left| \tilde{\mathcal{P}}(q)(2\mathcal{S}(q) - 1)\Delta f(q) \right| + \delta\epsilon$$

$$\leq \delta\epsilon \left( 1 + \frac{\tilde{\mathcal{P}}(D)\left|(2\mathcal{S}(q) - 1)\right|}{\min(\tilde{\mathcal{P}}(A), \tilde{\mathcal{P}}'(A))} + \frac{\tilde{\mathcal{P}}(D)\left|(2\mathcal{S}(q) - 1)\right|}{\min(\tilde{\mathcal{P}}(D), \tilde{\mathcal{P}}'(D))} \right)$$

$$\leq \delta\epsilon \left( 1 + \frac{\max(\mathcal{P}(D), \mathcal{P}'(D))}{\min(\mathcal{P}(A), \mathcal{P}'(A))} + \frac{\max(\mathcal{P}(D), \mathcal{P}'(D))}{\min(\mathcal{P}(D), \mathcal{P}'(D))} \right),$$

265  where the last inequality follows from the facts that $|(2\mathcal{S}(q) - 1)| \leq 1$, $\tilde{\mathcal{P}}(A)$ and $\tilde{\mathcal{P}}'(A)$ both lie
266  between $\mathcal{P}(A)$ and $\mathcal{P}'(A)$ and $\tilde{\mathcal{P}}(D)$ and $\tilde{\mathcal{P}}'(D)$ both lie between $\mathcal{P}(D)$ and $\mathcal{P}'(D)$.

267  In Appendix B.4, we derive a better upper bound on $|(2\mathcal{S}(q) - 1)|$ and derive the following:

$$\left| \text{Acc}(f_{\tilde{\mathcal{P}}}^{\text{DP}}, \tilde{\mathcal{P}}) - \text{Acc}(f_{\tilde{\mathcal{P}}'}^{\text{DP}}, \tilde{\mathcal{P}}') \right| \leq \delta\epsilon \left( 1 + \frac{\max(\mathcal{P}(A), \mathcal{P}'(A))}{\min(\mathcal{P}(A), \mathcal{P}'(A))} + \frac{\max(\mathcal{P}(A), \mathcal{P}'(A))}{\min(\mathcal{P}(D), \mathcal{P}'(D))} \right).$$

268  Putting the two upper bounds together yields the following:

$$\left| \text{Acc}(f_{\tilde{\mathcal{P}}}^{\text{DP}}, \tilde{\mathcal{P}}) - \text{Acc}(f_{\tilde{\mathcal{P}}'}^{\text{DP}}, \tilde{\mathcal{P}}') \right| \leq \delta\epsilon \left( 1 + \frac{\max(\mathcal{P}(A), \mathcal{P}'(A))}{\min(\mathcal{P}(A), \mathcal{P}'(A))} + \frac{\max(\mathcal{P}(D), \mathcal{P}'(D))}{\min(\mathcal{P}(D), \mathcal{P}'(D))} \right).$$

**Handling the equal-score and threshold events:** We now describe how to handle the two events.

1. Equal-score event: If the cell of $a$ has the same score as the adjacent cell lower in the sorted order, then we swap the two cells so that the cell of $a$ is lower in the order. Similarly, if the cell of $b$ has the same score as the adjacent cell higher in the order, then we swap the two cells so that the cell of $b$ is higher in the order. We update the classifier $f$ and note that this change has no impact on the accuracy of $f$.

2. Threshold event: The score of a merged cell containing $a$ or $b$ becomes exactly 0.5. We include the merged cell in the classifier $f$, again without changing accuracy.

Thus, in a sense, between any two occurrences of these events, the change in accuracy is bounded by an amount proportional to the mass transfer; when we reach these occurrences, the mass transfer is paused, the BOC changes without any change in accuracy. Furthermore, at every occurrence of the event, one of these three events happen: the cell containing $a$ moves down in the order, the cell containing $b$ moves up in the order, or an additional merged cell is placed above the threshold. Since the number of times these events can occur is upper bounded by the number of cells in the two groups, this process is finite. Therefore, adding over all the $\delta\epsilon$ mass transfers, we obtain the desired upper bound on the change in accuracy between the BOC's for $\mathcal{P}$ and $\mathcal{P}'$.

$$\epsilon\left(1 + \frac{\max\left(\mathcal{P}(A), \mathcal{P}'(A)\right)}{\min(\mathcal{P}(A), \mathcal{P}'(A))} + \frac{\max\left(\mathcal{P}(D), \mathcal{P}'(D)\right)}{\min(\mathcal{P}(D), \mathcal{P}'(D))}\right).$$

**From elementary to arbitrary:** Consider a general transition of distance $\epsilon$ from $\mathcal{P}$ to $\mathcal{P}'$. We invoke Lemma 1 to obtain intermediate distributions $\{\mathcal{P}_i\}$ with $TV(\mathcal{P}_{i-1}, \mathcal{P}_i) = \epsilon_i$ satisfying the decomposability and monotonocity properties. We apply the above proof for each elementary transition $\mathcal{P}_{i-1} \to \mathcal{P}_i$ of mass $\epsilon_i$. For accuracy, we derive

$$\begin{aligned}
\left|\text{Acc}(f_{\mathcal{P}}^{\text{DP}}, \mathcal{P}) - \text{Acc}(f_{\mathcal{P}'}^{\text{DP}}, \mathcal{P}')\right| &\leq \sum_i \left|\text{Acc}(f_{\mathcal{P}_{i-1}}^{\text{DP}}, \mathcal{P}_{i-1}) - \text{Acc}(f^{\text{DP}}\mathcal{P}_i, \mathcal{P}_i)\right| \\
&\leq \sum_i \epsilon_i \left(1 + \frac{\max(\mathcal{P}_{i-1}(A), \mathcal{P}_i(A))}{\min(\mathcal{P}_{i-1}(A), \mathcal{P}_i(A))} + \frac{\max(\mathcal{P}_{i-1}(D), \mathcal{P}_i(D))}{\min(\mathcal{P}_{i-1}(D), \mathcal{P}_i(D))}\right) \\
&\leq \sum_i \epsilon_i \left(1 + \frac{\max(\mathcal{P}(A), \mathcal{P}'(A))}{\min(\mathcal{P}(A), \mathcal{P}'(A))} + \frac{\max(\mathcal{P}(D), \mathcal{P}'(D))}{\min(\mathcal{P}(D), \mathcal{P}'(D))}\right) \\
&= \epsilon \left(1 + \frac{\max(\mathcal{P}(A), \mathcal{P}'(A))}{\min(\mathcal{P}(A), \mathcal{P}'(A))} + \frac{\max(\mathcal{P}(D), \mathcal{P}'(D))}{\min(\mathcal{P}(D), \mathcal{P}'(D))}\right),
\end{aligned}$$

where the third inequality follows from monotonocity and the last equation follows from decomposability. This completes the proof of the theorem. $\square$

We now state the following corollary, which follows from Claim 2 and Theorem 1. It roughly states that given 2 closeby distributions $\mathcal{P}, \mathcal{P}'$, the accuracy of the respective DP-Fair BOC's is similar on $\mathcal{P}$. Such a property is useful when we want a guarantee that intuitively says that if we train on the corrupted distribution $\mathcal{P}'$, we get a similar outcome to what we would have gotten had we trained on the true distribution $\mathcal{P}$.

**Corollary 1.** *Given distributions $\mathcal{P}, \mathcal{P}'$ with $TV(\mathcal{P}, \mathcal{P}') = \epsilon$, we have*

$$\left|Acc(f_{\mathcal{P}}^{DP}, \mathcal{P}) - Acc(f_{\mathcal{P}'}^{DP}, \mathcal{P})\right| \leq \epsilon\left(2 + \frac{\max\left(\mathcal{P}(A), \mathcal{P}'(A)\right)}{\min(\mathcal{P}(A), \mathcal{P}'(A))} + \frac{\max\left(\mathcal{P}(D), \mathcal{P}'(D)\right)}{\min(\mathcal{P}(D), \mathcal{P}'(D))}\right).$$

## 4 Equal Opportunity and Predictive Equality

Earlier, we presented results for Demographic Parity. Our results also extend to the popular fairness notions of Equal Opportunity and Predictive Equality [Hardt et al., 2016, Barocas et al., 2019]. We state the results here, and defer the proofs to Appendix C. We first define the fairness notions.

**Definition 7** (Equal TPR, or Equal Opportunity). Denote the true positive rate of $f$ on group $z$ by

$$\text{TPR}_z(f, \mathcal{P}) = \Pr[f(X, Z) = 1 \mid Y = 1, Z = z].$$

$f$ satisfies Equal Opportunity if the true positive rates are equal for both groups, i.e. $\text{TPR}_A(f, \mathcal{P}) = \text{TPR}_D(f, \mathcal{P})$. We quantify the unfairness of $f$ as the difference in true positive rates across groups , i.e.,

$$\text{Unf}_{\text{EO}}(f, \mathcal{P}) = |\text{TPR}_A(f, \mathcal{P}) - \text{TPR}_D(f, \mathcal{P})|.$$

**Definition 8** (Equal FPR, or Predictive Equality). Denote the false positive rate of of $f$ on group $z$ by

$$\text{FPR}_z(f, \mathcal{P}) = \Pr[f(X, Z) = 1 \mid Y = 0, Z = z].$$

$f$ satisfies Predictive Equality if the false positive rates are equal for both groups, i.e. $\text{FPR}_A(f, \mathcal{P}) = \text{FPR}_D(f, \mathcal{P})$. We quantify the unfairness of $f$ as the difference in false positive rates across groups , i.e.,

$$\text{Unf}_{\text{PE}}(f, \mathcal{P}) = |\text{FPR}_A(f, \mathcal{P}) - \text{FPR}_D(f, \mathcal{P})|.$$

*Remark.* Classifiers satisfying these notions of fairness will be referred to as EO-fair, and PE-fair respectively. The results for PE follow using the same proof techniques as that of EO (since we can just reverse the roles of the labels 0 and 1 in EO to get results for PE). We state the analogous results for PE in Appendix C.3. In addition, previous work has also considered equal False Negative rate (FNR) and equal True Negative rate (TNR) as notions of fairness. Obtaining equal TPR is equivalent to obtaining equal FNR, and obtaining equal TNR is equivalent to obtaining equal FPR, and hence results for these notions of fairness also follow.

We now state the results for EO.

**Claim 3** (EO Shift for a Fixed Hypothesis). *Given distributions $\mathcal{P}, \mathcal{P}'$, with $TV(\mathcal{P}, \mathcal{P}') \leq \epsilon$, and any hypothesis $f$, it holds that*

$$|Unf_{EO}(f, \mathcal{P}) - Unf_{EO}(f, \mathcal{P}')| \leq \epsilon \left( \frac{1}{\min(\mathcal{P}(A, 1), \mathcal{P}'(A, 1))} + \frac{1}{\min(\mathcal{P}(D, 1), \mathcal{P}'(D, 1))} \right),$$

*where $f_{\mathcal{P}}^{EO}, f_{\mathcal{P}'}^{EO}$ are the EO-Fair BOC's on $\mathcal{P}, \mathcal{P}'$ respectively.*

**Theorem 2** (Robustness of EO-Fair BOC). *Given distributions $\mathcal{P}, \mathcal{P}'$, such that $TV(\mathcal{P}, \mathcal{P}') = \epsilon$, we have that*

$$\left| Acc(f_{\mathcal{P}}^{EO}, \mathcal{P}) - Acc(f_{\mathcal{P}'}^{EO}, \mathcal{P}') \right| \leq \epsilon \left( 1 + 2\max(\mathcal{P}(1), \mathcal{P}'(1)) \left( \frac{1}{\min(\mathcal{P}(A, 1), \mathcal{P}'(A, 1))} + \frac{1}{\min(\mathcal{P}(D, 1), \mathcal{P}'(D, 1))} \right) \right),$$

*where $f_{\mathcal{P}}^{EO}, f_{\mathcal{P}'}^{EO}$ are the EO-Fair BOC's on $\mathcal{P}, \mathcal{P}'$ respectively.*

**Corollary 2.** *Given distributions $\mathcal{P}, \mathcal{P}'$, such that $TV(\mathcal{P}, \mathcal{P}') = \epsilon$, we have that*

$$\left| Acc(f_{\mathcal{P}}^{EO}, \mathcal{P}) - Acc(f_{\mathcal{P}'}^{EO}, \mathcal{P}) \right| \leq 2\epsilon \left( 1 + \max(\mathcal{P}(1), \mathcal{P}'(1)) \left( \frac{1}{\min(\mathcal{P}(A, 1), \mathcal{P}'(A, 1))} + \frac{1}{\min(\mathcal{P}(D, 1), \mathcal{P}'(D, 1))} \right) \right),$$

*where $f_{\mathcal{P}}^{EO}, f_{\mathcal{P}'}^{EO}$ are the EO-Fair BOC's on $\mathcal{P}, \mathcal{P}'$ respectively.*

## 5 Conclusion

Our findings collectively advance the theoretical understanding of fairness and robustness in adversarially noisy environments, providing a solid foundation for future research. Some directions for further work include extending our results for binary classification to multi-class classification, and regression. Another direction could be to look at relaxed or approximate versions of the fairness notions we considered. One could even look at other popular notions of fairness, or satisfying multiple fairness notions simultaneously. It would also be valuable to experimentally validate our theoretical claims. In addition, note that our results hold for adversarial noise, but it might be possible to strengthen the bounds if the noise came from a particular distribution.

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

## A   Comparison with Related Work

We now present detailed comparison with relevant previous work. In Blum et al. [2024], they aim to
avoid the non-robustness phenomena highlighted in Konstantinov and Lampert [2022], as follows.
Given any deterministic hypothesis class $\mathcal{H}$, and distributions $\mathcal{P}, \mathcal{P}'$ with $TV(\mathcal{P}, \mathcal{P}') = \epsilon$, they
construct a randomized closure of $\mathcal{H}$ called $PQ(\mathcal{H})$. Denote by $f, f'$ the optimal classifiers (subject
to DP constraints) on $\mathcal{P}, \mathcal{P}'$ restricted to $\mathcal{H}, PQ(\mathcal{H})$ respectively. They show that this satisfies a
one-directional Lipschitzness constraint, i.e., $\text{Acc}(f', \mathcal{P}') \geq \text{Acc}(f, \mathcal{P}) - O(\epsilon)$. They also show
analogous results for EO and PE. Our setup has some key differences. We do not consider any
arbitrary $\mathcal{H}$, but the BOC setting which includes all deterministic classifiers (and the 1-skeleton of
their convex closure). More crucially, our robustness guarantee is stronger, as their Lipschitzness
guarantee is only one-directional. In addition, in most cases, their output hypothesis incorporates
a lot of randomness, outputting a randomized decision on all elements in the domain, whereas our
output hypothesis is randomized on at most one element.

In the concurrent work of Xian and Zhao [2024], the sensitivity analysis (Theorem 3.1) bounds the
drop in accuracy of the optimal fair classifier under a shift in distribution, for the multiclass and

multigroup setting, focusing on continuous domains. However, their sensitivity analysis only holds for either a shift in the label distribution, or in the group membership distribution, whereas our robustness guarantee works for adversarial distribution shifts. Adversarial or arbitrary distribution shifts are strictly more general than label/covariate shifts, and moreover, they cannot be simulated by any combination of label/covariate shifts. In addition, in their sensitivity analysis (Theorem 3.1, 2nd result), the change in accuracy due to group distribution shift, is a constant independent of the amount of distribution shift (in the case of perfect fairness). We prove a stronger Lipschitzness guarantee, where the excess risk goes to 0 as distance between the distributions becomes arbitrarily small. Furthermore, they do not provide a description of the Randomized Fair BOC in the case of discrete domains, whereas we provide a complete characterization of the same, show that it is minimally random. In addition, our algorithm (to output the Fair BOC on a distribution) is very simple and efficient, running in $O(|\mathcal{X}|\log(|\mathcal{X}|))$ time, while their algorithm solves a large linear program with $O(|\mathcal{X}|)$ constraints in $O(|\mathcal{X}|)$ variables, requiring a much higher complexity.

Chen et al. [2024] contains a similar sensitivity analysis as Xian and Zhao [2024], for the same setting except binary group and binary class. Unlike us, they do not deal with adversarial distributions shifts, but only label distribution shifts and/or group distribution shifts. In addition, our setups are fundamentally different, theirs being the continuous case, and ours being the discrete case. Moreover, their sensitivity analysis (Theorem 2) is looser, and has an extra additive error term, unlike ours and that of Xian and Zhao [2024]. Besides, they do not deal with the case of perfect fairness, and require $\delta > 0$. Chen et al. [2022] also consider fairness under distribution shift. Their result is fundamentally different, and essentially shows that the fairness of a fixed hypothesis class on two similar distributions is similar. This is essentially what we show in Claims 2/3/7, however, they only deal with label and covariate shifts, while we tackle the more general case of adversarial distribution shifts.

# B  Missing Results from Section 3

## B.1  Non-Robustness of the Deterministic Fair BOC (approximate fairness)

We show through the example below that the non-robustness phenomenon highlighted in Claim 1 also holds when we only require approximate fairness[7]. In particular, this can hold in the case where sensitive group populations are highly imbalanced, for example when the mass of group $A$ is much larger than the mass of group $D$, i.e., $P(A) \gg P(D)$. We set $\delta = 0.25$, and slightly modify the example in Claim 1, where we skew the probability mass towards Group $A$ (in Claim 1, the group masses are balanced).

**Claim 4** (Non-Robustness of Deterministic Fair BOC (approximate fairness)). *There exist distributions $\mathcal{P}, \mathcal{P}'$ with $TV(\mathcal{P}, \mathcal{P}') = \epsilon$, such that the deterministic DP-Fair BOC's $f, f'$ on $\mathcal{P}, \mathcal{P}'$, respectively, satisfies $|Acc(f, \mathcal{P}) - Acc(f', \mathcal{P}')| \geq \Omega(1)$.*

*Proof.* Consider a distribution P, where $P, S(x_1, A) = (0.4, 1)$ — $P, S(x_1, D) = (0.1, 1)$ — $P, S(x_2, A) = (0.4, 0)$ — $P, S(x_2, D) = (0.1, 0)$ Consider the (deterministic) classifier $f$, with $f(x_1, A) = f(x_1, D) = 1, f(x_2, A) = f(x_2, D) = 0$. $f$ satisfies DP, and Acc$(f) = 1$. Consider the neighboring distribution $P'$ differing only on $(x_1, D), (x_2, D)$, as follows.

$P', S(x_1, D) = (0.1 + 0.05, 1)$ — $P', S(x_2, D) = (0.1 - 0.05, 0)$

If we apply $f$ on $P'$, it does not satisfy approximate DP for any $\delta < 0.25$, even though $TV(P, P')$ is small (0.05). There are only 2 deterministic classifiers satisfying approximate DP for any $\delta < 0.25$, either the constant 1 classifier $f_1$, or the constant 0 classifier $f_0$, with Acc$(f_1) = 1/2 + 0.05$, and Acc$(f_0) = 1/2 - 0.05$. Hence, the difference in accuracy of the deterministic (approximate) DP-Fair BOC on closeby $P, P'$ is almost 0.5, demonstrating non-robustness. □

## B.2  NP-Completeness of Deterministic Fair Bayes Optimal Classifiers

**Claim 5** (NP-Completeness of Deterministic DP-Fair BOC). *Given a distribution $\mathcal{P}$, the problem of computing the deterministic DP-Fair BOC is NP-complete.*

---

[7]We define a $\delta$-approximately fair classifier as follows, "If $r$ denotes selection rate, a classifier $f$ is $\delta$-approximately DP-fair if $|r(f, A) - r(f, D)| < \delta$".

*Proof.* We formalize the deterministic DP-Fair BOC decision problem as follows: Given a probability distribution $\mathcal{P}$ and a score function $\mathcal{S}$ over a domain $\mathcal{X} \times \mathcal{Z}$ and an accuracy $\alpha$, determine whether there exists a deterministic fair classifier with accuracy at least $\alpha$.

It is easy to see that the above problem is is in NP since one can guess the 0-1 classification for each item in the domain and check in polynomial time that the resulting classifier is fair and satisfies the accuracy bound by verifying two linear inequalities. We now show that the problem is NP-hard via a polynomial-time reduction from the NP-complete Partition problem, which we state below.

Partition problem: Given a set $S$ of $n$ positive integers $a_1, a_2, \ldots, a_n$ summing to $2s$, determine whether there exists a subset of $S$ that sums to $s$.

The reduction from Partition to the deterministic DP-Fair BOC problem is as follows. Given an instance $I$ of Partition, we create an instance $I'$ of the deterministic DP-fair BOC problem. Instance $I'$ has $n + 2$ items—$(x_1, A)$ with mass $1/4$ and score $1$, item $(x_2, A)$ with mass $1/4$ and score $0$, and then $n$ items $(y_i, D)$ with mass $a_i/(4s)$ and score $0.5$—and ask whether there is a deterministic DP-Fair classifier with accuracy $\alpha \geq 3/4$. It is clear that there are at most 3 kinds of deterministic DP-fair classifiers: (i) the all-0 classifier that classifies all items as 0, (ii) the all-1 classifier that classifies all items as 1, and (iii) if and only f $I$ is a yes-instance with $S$ partitioned into $S_1$ and $S_2$ of equal sums, then the classifier that accepts exactly one of $(x_1, A)$ or $(x_2, A)$ and accepts all items in $(y_i, D)$ with $a_i \in S_1$ and rejecting all items in $(y_i, D)$ with $a_i \in S_2$. The first two classifiers have accuracy $1/2$ while the third, if it exists, has accuracy $3/4$ if $(x_1, A)$ is accepted and less than $3/4$ otherwise. Thus, there exists a deterministic Fair BOC for instance $I'$ with $3/4$ accuracy if and only if $I$ is a yes-instance for the Partition problem. Clearly, the reduction is of time polynomial in the size of the deterministic DP-Fair BOC instance, thus establishing its NP-completeness.

Since determining the existence of a deterministic DP-fair classifier with accuracy at least $3/4$ is NP-complete it follows immediately that finding a deterministic DP-Fair BOC is also NP-complete. □

## B.3 Maximal Accuracy Gain for Randomized Classifiers

Consider the example in Claim 1, and consider the following randomized classifier $f'$, where

$$f'(x_1, A) = f'(x_1, D) = 1, f'(x_2, A) = 4\epsilon, f'(x_2, D) = 0.$$

It is easy to see that $f'$ satisfies DP on $\mathcal{P}'$, and $\text{Acc}(f') = 1 - \epsilon$. Hence, the randomized DP-fair BOC improves over the accuracy of its deterministic counterpart by $0.5 - 2\epsilon$, where $\epsilon > 0$ can be made arbitrarily small (so the gain in accuracy approaches $0.5$). In the following claim, we argue that this example is tight, i.e., we cannot hope to achieve an improvement over $0.5$.

**Claim 6** (Bound in Accuracy Gain for Randomized classifiers). *Given any distribution $\mathcal{P}$, the difference in accuracy of the Randomized and Deterministic DP-Fair BOC's on $\mathcal{P}$ is strictly lesser than $0.5$.*

*Proof.* Note that the constant classifiers $f_0, f_1$ always satisfy DP, and $\text{Acc}(f_0) = 1 - \text{Acc}(f_1)$. Hence, the minimum accuracy of the optimal DP-fair deterministic classifier is $0.5$. The maximum accuracy of its randomized counterpart is bounded by $1$, hence bounding the difference in accuracy by $0.5$. It suffices to show that these 2 events cannot occur simultaneously. Note that if some classifier has perfect accuracy, then all cells in the domain have score of either 0 or 1. In particular, this also holds if the optimal DP-fair randomized classifier has accuracy $1$. However, observe that if we randomize over any cell with score of $0(1)$, we are accepting (rejecting) a part of it, leading to a loss in accuracy. This implies that any classifier with accuracy $1$ has to be deterministic, concluding our proof. □

## B.4 Completion of the Robustness Analysis for Demographic Parity

In this section, we present the argument that was deferred in the proof of Theorem 1. This argument concerns a better upper bound on $|2\mathcal{S}(q) - 1|$ than the vacuous bound of 1, where $q$ is the element that is split by the threshold corresponding to the classifier $f$. Notice that since by assumption, $f$ splits $q$ in the middle, we know that there is a portion of $q$ that is rejected. Hence, the weighted score of a merged cell involving $q$ (say $C_q$) has score below the threshold of $0.5$. Let $C_q$ contain some element $t$ from group $A$. We are able to bound the score of $\mathcal{S}(q)$ by the following chain of

inequalities.

$$\mathcal{S}(\mathcal{C}_q) \leq 0.5 \implies \mathcal{S}(q)\mathcal{P}(D) + \mathcal{S}(t)\mathcal{P}(A) \leq 0.5\left(\mathcal{P}(D) + \mathcal{P}(A)\right)$$
$$\implies \mathcal{S}(q)\mathcal{P}(D) \leq 0.5\left(\mathcal{P}(D) + \mathcal{P}(A)\right)$$
$$\implies 2\mathcal{S}(q) - 1 \leq \frac{\mathcal{P}(A)}{\mathcal{P}(D)} \tag{1}$$

Since $f$ splits $q$ in the middle, there is also a portion of $q$ that is accepted. Hence, the weighted score of a merged cell involving $q$ (say $\mathcal{C}_q$) has score above the threshold of $0.5$. Let $C_q$ contain some element $t$ from group $A$. We are able to bound the score of $\mathcal{S}(q)$ by the following chain of inequalities.

$$\mathcal{S}(\mathcal{C}_q) \geq 0.5 \implies \mathcal{S}(q)\mathcal{P}(D) + \mathcal{S}(t)\mathcal{P}(A) \geq 0.5\left(\mathcal{P}(D) + \mathcal{P}(A)\right)$$
$$\implies \mathcal{S}(q)\mathcal{P}(D) + \mathcal{P}(A) \geq 0.5\left(\mathcal{P}(D) + \mathcal{P}(A)\right)$$
$$\implies \mathcal{S}(q)\mathcal{P}(D) \geq 0.5\left(\mathcal{P}(D) - \mathcal{P}(A)\right)$$
$$\implies 2\mathcal{S}(q) - 1 \geq -\frac{\mathcal{P}(A)}{\mathcal{P}(D)} \tag{2}$$

Combining Equations 1 and 2, we get

$$|2\mathcal{S}(q) - 1| \leq \frac{\mathcal{P}(A)}{\mathcal{P}(D)} \tag{3}$$

Using Equation 3, we get that

$$\left|\mathrm{Acc}(f_{\tilde{\mathcal{P}}}^{\mathrm{DP}}, \tilde{\mathcal{P}}) - \mathrm{Acc}(f_{\tilde{\mathcal{P}}'}^{\mathrm{DP}}, \tilde{\mathcal{P}}')\right| \leq \delta\epsilon\left(\frac{1}{\min(\tilde{\mathcal{P}}(A), \tilde{\mathcal{P}}'(A))} + \frac{1}{\min(\tilde{\mathcal{P}}(D), \tilde{\mathcal{P}}'(D))}\right)\tilde{\mathcal{P}}(D)\frac{\tilde{\mathcal{P}}(A)}{\tilde{\mathcal{P}}(D)} + \delta\epsilon$$
$$= \delta\epsilon\left(\frac{1}{\min(\tilde{\mathcal{P}}(A), \tilde{\mathcal{P}}'(A))} + \frac{1}{\min(\tilde{\mathcal{P}}(D), \tilde{\mathcal{P}}'(D))}\right)\tilde{\mathcal{P}}(A) + \delta\epsilon$$
$$\leq \delta\epsilon\left(1 + \frac{\max\left(\mathcal{P}(A), \mathcal{P}'(A)\right)}{\min(\mathcal{P}(A), \mathcal{P}'(A))} + \frac{\max\left(\mathcal{P}(A), \mathcal{P}'(A)\right)}{\min(\mathcal{P}(D), \mathcal{P}'(D))}\right) \tag{4}$$

The last equation follows by monotonicity. This completes the missing argument in the proof of Theorem 1.

## B.5 Proof of Lemma 1

*Proof.* We will prove the desired claim for $n$ equal to number of elements $q$ for which $\mathcal{P}(q) \neq \mathcal{P}(q')$. Our proof is by induction on $n$. For the base case, we have $n = 0$, in which case $\mathcal{P} = \mathcal{P}'$ and the claim trivially holds. For the induction step, let $a$ be an element such that $\mathcal{P}(a) \neq \mathcal{P}'(a)$. Suppose $\mathcal{P}(a) > \mathcal{P}'(a)$ and $a$ is in group $A$; the arguments for the other scenarios are analogous. We consider two cases. The first case is when there exists $b \in A$ such that $\mathcal{P}(b) < \mathcal{P}'(b)$. We define $\tilde{\mathcal{P}}$ as the same as $\mathcal{P}$ except that

$$\tilde{\mathcal{P}}(a) = \mathcal{P}(a) - \min\{\mathcal{P}(a) - \mathcal{P}'(a), \mathcal{P}'(b) - \mathcal{P}(b)\}$$
$$\tilde{\mathcal{P}}(b) = \mathcal{P}(b) + \min\{\mathcal{P}(a) - \mathcal{P}'(a), \mathcal{P}(b) - \mathcal{P}'(b)\}.$$

Note that either $\tilde{\mathcal{P}}(a) = \mathcal{P}'(a)$ or $\tilde{\mathcal{P}}(b) = \mathcal{P}'(b)$, which implies that the number of elements for which $\tilde{\mathcal{P}}$ and $\mathcal{P}'$ differ is less than $n$. Furthermore, $\mathcal{P}(A) = \tilde{\mathcal{P}}(A)$ and $\mathcal{P}(D) = \tilde{\mathcal{P}}(D)$. By induction, there exist a sequence of $m < n$ distributions $\tilde{\mathcal{P}} = \mathcal{P}_0, \mathcal{P}_1, \ldots, \mathcal{P}_m = \mathcal{P}'$ satisfying the decomposability and monotonicity properties. Appending the elementary transition $\mathcal{P} \to \tilde{\mathcal{P}}$ to the above sequence yields the desired sequence for $\mathcal{P}$ and $\mathcal{P}'$ with the decomposability and monotonicity properties.

The second case is when there does not exist any $b \in A$ such that $\mathcal{P}(b) < \mathcal{P}'(b)$. So, we have $\mathcal{P}(A) > \mathcal{P}'(A)$. Furthermore, there exists $b \in D$ such that $\mathcal{P}(b) < \mathcal{P}'(b)$. We define $\tilde{\mathcal{P}}$ in the same way as for the first case. Again, we have that either $\tilde{\mathcal{P}}(a) = \mathcal{P}'(a)$ or $\tilde{\mathcal{P}}(b) = \mathcal{P}'(b)$, which implies that the number of elements for which $\tilde{\mathcal{P}}$ and $\mathcal{P}'$ differ is less than $n$. Furthermore, $\mathcal{P}(A) > \tilde{\mathcal{P}}(A)$ and $\mathcal{P}(D) < \tilde{\mathcal{P}}(D)$. By induction, there exist a sequence of $m < n$ distributions $\tilde{\mathcal{P}} = \mathcal{P}_0, \mathcal{P}_1, \ldots, \mathcal{P}_m = \mathcal{P}'$ satisfying the decomposability and monotonicity properties. Again, appending the elementary transition $\mathcal{P} \to \tilde{\mathcal{P}}$ to the above sequence yields the desired sequence for $\mathcal{P}$ and $\mathcal{P}'$ with the decomposability and monotonicity properties. $\qquad\square$

## B.6 Proof of Claim 2

*Proof.* We first establish the desired statements for the special case where the transition from $\mathcal{P}$ to $\mathcal{P}'$ is elementary in that the only difference between the two distributions is that there are two elements $a$ and $b$ that have $\epsilon$ more mass and $\epsilon$ less mass, respectively, in $\mathcal{P}$ as compared to $\mathcal{P}'$ (all other elements have the same mass in the two distributions). At the end, we invoke Lemma 1 and transitivity to establish the general claim.

**Accuracy:** Divide the domain into 4 parts based on whether a point falls in categories TP, FP, TN, or FN according to $f$. Denote the probability mass of elements in category $E$ under $f$ by $\mathcal{P}(E)$. We know that $\mathrm{Acc}(f, \mathcal{P}) = \mathcal{P}(\mathrm{TP} \cup \mathrm{TN})$. Doing a simple case by case analysis, we observe that in the worst case, $a$ belongs to $\mathrm{TP} \cup \mathrm{TN}$, and $b$ belongs to $\mathrm{FP} \cup \mathrm{FN}$. This transition leads to a loss in accuracy of $\epsilon$, i.e., $\mathrm{Acc}(f, \mathcal{P}') = \mathrm{Acc}(f, \mathcal{P}) - \epsilon$. We note that it is enough to consider a loss in accuracy, since we can reverse the roles of the distributions and use the same argument for gain as that for loss.

**Demographic Parity:** First we notice the following

$$
\begin{aligned}
|\mathrm{Unf}_{\mathrm{DP}}(f, \mathcal{P}) - \mathrm{Unf}_{\mathrm{DP}}(f, \mathcal{P}')| &= ||\mathrm{r}_A(f, \mathcal{P}) - \mathrm{r}_D(f, \mathcal{P})| - |\mathrm{r}_A(f, \mathcal{P}') - \mathrm{r}_D(f, \mathcal{P}')|| \\
&\leq |(\mathrm{r}_A(f, \mathcal{P}) - \mathrm{r}_D(f, \mathcal{P})) - (\mathrm{r}_A(f, \mathcal{P}') - \mathrm{r}_D(f, \mathcal{P}'))| \\
&\qquad\qquad\qquad\qquad\qquad\qquad\text{(Triangle inequality)} \\
&= |(\mathrm{r}_A(f, \mathcal{P}) - \mathrm{r}_A(f, \mathcal{P}')) + (\mathrm{r}_D(f, \mathcal{P}') - \mathrm{r}_D(f, \mathcal{P}))| \\
&\leq |\mathrm{r}_A(f, \mathcal{P}) - \mathrm{r}_A(f, \mathcal{P}')| + |\mathrm{r}_D(f, \mathcal{P}') - \mathrm{r}_D(f, \mathcal{P})| \qquad (5)
\end{aligned}
$$

The above argument breaks up the change in unfairness into two terms: (i) $\Delta \mathrm{r}_A \triangleq |\mathrm{r}_A(f, \mathcal{P}) - \mathrm{r}_A(f, \mathcal{P}')|$, which is the difference in selection rates of $f$ for $\mathcal{P}$, and $\mathcal{P}'$ on $A$ and (ii) $\Delta \mathrm{r}_D \triangleq |\mathrm{r}_D(f, \mathcal{P}) - \mathrm{r}_D(f, \mathcal{P}')|$, which is the difference in selection rates of $f$ for $\mathcal{P}$ and $\mathcal{P}'$ on $D$.

We proceed to bound $\Delta \mathrm{r}_A$, and an identical argument can be used to bound $\Delta \mathrm{r}_D$. In our argument, we divide the domain into 4 parts based on the group membership and labeling according to $f$. Let the probability mass of elements in group $z$ with label $y$ under classifier $f$ be denoted by $\mathcal{P}(z, f_y)$. If $a, b$ lie in the same group $z$ then $\mathcal{P}(A)$ remains unchanged, and it is easy to see that the maximum value of $\Delta \mathrm{r}_A$ is $\frac{\epsilon}{\mathcal{P}(A)}$, when $\mathcal{P}'(A, f_1) = \mathcal{P}(A, f_1) \pm \epsilon$. In case $a \in A$, and $b \in D$, then $\mathcal{P}'(A) = \mathcal{P}(A) - \epsilon$. We know that $\mathcal{P}'(A) = \mathcal{P}'(A, f_1) + \mathcal{P}'(A, f_0)$. Either $a$ lies completely in $(A, f_1)$, completely in $(A, f_0)$, or in both (if we are randomizing over the cell containing $a$). We first consider the first case, where $\mathcal{P}'(A, f_1) = \mathcal{P}(A, f_1) - \epsilon$.

$$
\begin{aligned}
|\mathrm{r}_A(f, \mathcal{P}) - \mathrm{r}_A(f, \mathcal{P}')| &= \left| \frac{\mathcal{P}(A, f_1)}{\mathcal{P}(A)} - \frac{\mathcal{P}(A, f_1) - \epsilon}{\mathcal{P}(A) - \epsilon} \right| \\
&= \left| \frac{\mathcal{P}(A)\epsilon - \mathcal{P}(A, f_1)\epsilon}{\mathcal{P}(A)\,(\mathcal{P}(A) - \epsilon)} \right| \\
&\leq \epsilon \left| \frac{1}{\mathcal{P}(A) - \epsilon} \right| \\
&\leq \epsilon \left( \frac{1}{\min(\mathcal{P}(A), \mathcal{P}'(A))} \right)
\end{aligned}
$$

We now consider the second case, where $\mathcal{P}'(A, f_0) = \mathcal{P}(A, f_0) - \epsilon$.

$$
\begin{aligned}
|\mathrm{r}_A(f, \mathcal{P}) - \mathrm{r}_A(f, \mathcal{P}')| &= \left| \frac{\mathcal{P}(A, f_1)}{\mathcal{P}(A)} - \frac{\mathcal{P}(A, f_1)}{\mathcal{P}(A) - \epsilon} \right| \\
&= \left| \frac{\mathcal{P}(A, f_1)\epsilon}{\mathcal{P}(A)\,(\mathcal{P}(A) - \epsilon)} \right| \\
&\leq \epsilon \left| \frac{1}{\mathcal{P}(A) - \epsilon} \right| \\
&\leq \epsilon \left( \frac{1}{\min(\mathcal{P}(A), \mathcal{P}'(A))} \right)
\end{aligned}
$$

582 It is easy to see that in the third case, where $a$ lies in both $(A, f_1)$ and $(A, f_0)$, $\Delta \mathrm{r}_A$ is bounded by
583 the max value of $\Delta \mathrm{r}_A$ of cases 1 and 2.

584 Here we argued for when $A$ loses mass. Using symmetry, we can similarly argue the case where $A$
585 gains mass, i.e., $a \in D$, and $b \in A$, leading to $\mathcal{P}'(A) = \mathcal{P}(A) + \epsilon$. Hence, we conclude that

$$|\mathrm{r}_A(f, \mathcal{P}) - \mathrm{r}_A(f, \mathcal{P}')| \leq \epsilon \left( \frac{1}{\min(\mathcal{P}(A), \mathcal{P}'(A))} \right) \tag{6}$$

586 Also, here we argued for group $A$, and an identical argument for $D$ shows that

$$|\mathrm{r}_D(f, \mathcal{P}) - \mathrm{r}_D(f, \mathcal{P}')| \leq \epsilon \left( \frac{1}{\min(\mathcal{P}(D), \mathcal{P}'(D))} \right) \tag{7}$$

587 Plugging Equations 6 and 7 into Equation 5, we get that

$$|\mathrm{Unf}_{\mathrm{DP}}(f, \mathcal{P}) - \mathrm{Unf}_{\mathrm{DP}}(f, \mathcal{P}')| \leq$$
$$\epsilon \left( \frac{1}{\min(\mathcal{P}(A), \mathcal{P}'(A))} \right) + \epsilon \left( \frac{1}{\min(\mathcal{P}(D), \mathcal{P}'(D))} \right)$$

588 **From elementary to arbitrary:** Consider a general transition of distance $\epsilon$ from $\mathcal{P}$ to $\mathcal{P}'$. We
589 invoke Lemma 1 to obtain intermediate distributions $\{\mathcal{P}_i\}$ with $TV(\mathcal{P}_{i-1}, \mathcal{P}_i) = \epsilon_i$ satisfying the
590 decomposability and monotonocity properties. We apply the above proof for each elementary tran-
591 sition $\mathcal{P}_{i-1} \to \mathcal{P}_i$ of mass $\epsilon_i$. For accuracy, we derive

$$|\mathrm{Acc}(f, \mathcal{P}) - \mathrm{Acc}(f, \mathcal{P}')| \leq \sum_i |\mathrm{Acc}(f, \mathcal{P}_{i-1}) - \mathrm{Acc}(f, \mathcal{P}_i)|$$
$$\leq \sum_i \epsilon_i$$
$$= \epsilon.$$

592 For Demographic Parity, we derive

$$|\mathrm{Unf}_{\mathrm{DP}}(f, \mathcal{P}) - \mathrm{Unf}_{\mathrm{DP}}(f, \mathcal{P}')| \leq \sum_i |\mathrm{Unf}_{\mathrm{DP}}(f, \mathcal{P}_{i-1}) - \mathrm{Unf}_{\mathrm{DP}}(f, \mathcal{P}_i)|$$
$$\leq \sum_i \epsilon_i \left( \frac{1}{\min(\mathcal{P}_{i-1}(A), \mathcal{P}_i(A))} + \frac{1}{\min(\mathcal{P}_{i-1}(D), \mathcal{P}_i(D))} \right)$$
$$\leq \sum_i \epsilon_i \left( \frac{1}{\min(\mathcal{P}(A), \mathcal{P}'(A))} + \frac{1}{\min(\mathcal{P}(D), \mathcal{P}'(D))} \right)$$
$$= \epsilon \left( \frac{1}{\min(\mathcal{P}(A), \mathcal{P}'(A))} + \frac{1}{\min(\mathcal{P}(D), \mathcal{P}'(D))} \right),$$

593 where the second inequality follows from monotonocity and the last equation follows from decom-
594 posability. This completes the proof of the claim. $\qquad\square$

595 **B.7 Proof of Lemma 2**

596 *Proof.* The contribution to accuracy of an element $q$ is given by

$$\mathcal{P}(q)\mathcal{S}(q)f(q) + \mathcal{P}(q)(1 - \mathcal{S}(q))(1 - f(q)) = 2\mathcal{P}(q)\mathcal{S}(q)f(q) + \mathcal{P}(q) - \mathcal{P}(q)\mathcal{S}(q) - \mathcal{P}(q)f(q)$$

597 If $\mathcal{P}$ changes by $\epsilon$ and $f(q)$ changes by $\Delta f(q)$ (and remains constant elsewhere), then we can split
598 the process into two parts: (i) $f(q)$ changes by $\Delta f(q)$ (and remains constant elsewhere) while $\mathcal{P}$
599 remains constant, and (ii) $\mathcal{P}$ changes by $\epsilon$ while $f$ remains constant. We consider each of the parts.

600 If $\mathcal{P}$ remains fixed, and $f(q)$ changes by $\Delta f(q)$ to give $f'(q)$, then change in accuracy on $q$ (and
601 also overall accuracy) is given by

$$|2\mathcal{P}(q)\mathcal{S}(q)\Delta f(q) - \mathcal{P}(q)\Delta f(q)| = |\mathcal{P}(q)(2\mathcal{S}(q) - 1)\Delta f(q)|$$

602 If $\mathcal{P}$ changes by $\epsilon$, and $f'$ remains constant, then by Claim 2 the change in accuracy is bounded by
603 $\epsilon$. Thus, the total change in accuracy is bounded as follows.

$$|\mathrm{Acc}(f, \mathcal{P}) - \mathrm{Acc}(f', \mathcal{P}')| \leq |\mathcal{P}(q)(2\mathcal{S}(q) - 1)\Delta f(q)| + \epsilon.$$

604 $\qquad\square$

## C  Equal Opportunity and Predictive Equality (continued)

### C.1  Fair Bayes Optimal Classifier

When discussing the DP-Fair BOC, we considered mass-threshold classifiers $\tilde{\mathcal{T}}_t$, that select $\mathcal{C}_z(t)$, and reject $\mathcal{C}_z - \mathcal{C}_z(t)$, for both $z = A$, and $z = D$. $\tilde{\mathcal{T}}_t$ applies the same threshold $t$ to both groups $A$ and $D$. In this section, we consider groupwise mass-threshold classifiers $\tilde{\mathcal{T}}_{t_A, t_D}$ that apply different thresholds $t_A$ and $t_D$ to groups $A$ and $D$ respectively.

Denote the True Positive rate of a classifier $f$ restricted to a cell $\mathcal{C}$ by $\mathrm{TPR}(f(\mathcal{C}))$. Given $r \in (0, 1]$, there is a unique classifier $\tilde{\mathcal{T}}_{t_A, t_D}$, such that $\mathrm{TPR}(\tilde{\mathcal{T}}_{t_A, t_D}(\mathcal{C}_A)) = \mathrm{TPR}(\tilde{\mathcal{T}}_{t_A, t_D}(\mathcal{C}_D)) = r$. Denote this classifier by $f_r$. Given $r = 0$, $\tilde{\mathcal{T}}_{t_A, t_D}$ need not be unique as there could exist cells with score 1. In that case, we define $f_0$ to be the unique groupwise mass-threshold classifier accepting exactly the cells with score 1. Denote the groupwise thresholds of $f_r$ by $r^A$ and $r^D$ respectively, i.e., $f_r = \tilde{\mathcal{T}}_{r^A, r^D}$. We now introduce some terminology, before detailing the EO-Fair BOC as characterized in Agarwal and Deshpande [2022].

**Definition 9** (TP-Boundaries). Recall the set of score-boundaries $\mathcal{I}$. We then define the set of TP-boundaries $\mathcal{I}_{TP}$ as
$$\mathcal{I}_{TP} = \{r \mid r^A \in \mathcal{I}, \text{ or } r^D \in \mathcal{I}\},$$
$\mathcal{I}_{TP}$ essentially consists of all the true positive rates $r$, such that, the corresponding groupwise threshold classifier $f_r = \tilde{\mathcal{T}}_{r^A, r^D}$ has a threshold at a point in the set of score boundaries $\mathcal{I}$.

As with DP, we define the notion of a merged cell, but notice that it differs from the notion of merged cell in the case of DP.

**Definition 10** (Merged cell (EO)). Consider $r_i \in \mathcal{I}_{TP}$, and define a merged cell $\mathcal{C}_i$, where
$$\mathcal{C}_i = \mathcal{A}(f_{r_i}) - \mathcal{A}(f_{r_{i-}}),$$
where $r_{i-}$ denotes the element in $\mathcal{I}_{TP}$ preceding $r_i$.

**Characterization**  Given a distribution $\mathcal{P}$ over $\mathcal{X} \times \mathcal{Z} \times \mathcal{Y}$, the EO-Fair BOC $f_{\mathcal{P}}^{\mathrm{EO}}$ is given by the mass-threshold classifier is given by the group wise mass-threshold classifier $f_{r'}$, where $r' = r_i \in \mathcal{I}$ is the unique $i$ such that $\mathcal{S}(C_i) \geq 0.5$, and $\mathcal{S}(C_{i+}) < 0.5$, where $r_{i+}$ denotes the element in $\mathcal{I}_{TP}$ after $r_i$.

### C.2  Robustness to Adversarial Distribution Shift

We study the robustness of the EO-Fair BOC to adversarial distribution shift. We show that given two similar distributions $\mathcal{P}, \mathcal{P}'$, the accuracy of the EO-Fair BOC on the respective distributions is similar (satisfies local Lipschitzness). Before proving the main result (Theorem 2), we prove Claim 3, which analyzes the change in unfairness, with respect to EO, of a fixed classifier due to a distribution shift. Such a property is useful when we want a guarantee that if we train a classifier on the corrupted distribution $\mathcal{P}'$, the performance of the classifier on the actual distribution $\mathcal{P}$ will be similar to that on $\mathcal{P}'$.

*Proof of Claim 3.* As in the proof of Claim 2, it follows from Lemma 1 and transitivity that it is enough to prove the statement of the claim for elementary transitions. We consider a transition $a \to b$ of mass $\epsilon$. We first derive

$$
\begin{aligned}
|\mathrm{Unf}_{\mathrm{EO}}(f, \mathcal{P}) - \mathrm{Unf}_{\mathrm{EO}}(f, \mathcal{P}')| &= ||\mathrm{TPR}_A(f, \mathcal{P}) - \mathrm{TPR}_D(f, \mathcal{P})| - |\mathrm{TPR}_A(f, \mathcal{P}') - \mathrm{TPR}_D(f, \mathcal{P}')|| \\
&\leq |(\mathrm{TPR}_A(f, \mathcal{P}) - \mathrm{TPR}_D(f, \mathcal{P})) - (\mathrm{TPR}_A(f, \mathcal{P}') - \mathrm{TPR}_D(f, \mathcal{P}'))| \\
&\qquad\qquad\qquad\qquad\qquad\qquad\qquad\qquad\qquad \text{(Triangle inequality)} \\
&= |(\mathrm{TPR}_A(f, \mathcal{P}) - \mathrm{TPR}_A(f, \mathcal{P}')) + (\mathrm{TPR}_D(f, \mathcal{P}') - \mathrm{TPR}_D(f, \mathcal{P}))| \\
&\leq |\mathrm{TPR}_A(f, \mathcal{P}) - \mathrm{TPR}_A(f, \mathcal{P}')| + |\mathrm{TPR}_D(f, \mathcal{P}') - \mathrm{TPR}_D(f, \mathcal{P})|
\end{aligned}
\tag{8}
$$

This breaks up the change in unfairness into two terms, which correspond to the difference in true positive rates of $f$ for $\mathcal{P}$ and $\mathcal{P}'$ on $A, D$ respectively (denoted by $\Delta\mathrm{TPR}_A, \Delta\mathrm{TPR}_D$). Divide the

643 domain into 8 parts based on the group membership and whether a point falls in TP, FP, TN, or FN
644 according to $f$. Denote the probability mass of elements in group $z$ in category $E$ under $f$ by $\mathcal{P}(E_z)$.
645 We know that $\mathcal{P}(A) = \mathcal{P}(A, 1) + \mathcal{P}(A, 0) = (\mathcal{P}(\text{TP}_A) + \mathcal{P}(\text{FN}_A)) + (\mathcal{P}(\text{TN}_A) + \mathcal{P}(\text{FP}_A))$.

646 We proceed to bound $\Delta\text{TPR}_A$, and an identical argument can be used to bound $\Delta\text{TPR}_D$. If $a, b$ lie
647 in $\mathcal{P}(A, 1)$, it remains unchanged, and it is easy to see that the maximum value of $\Delta\text{TPR}_A$ is $\frac{\epsilon}{\mathcal{P}(A,1)}$,
648 when $\mathcal{P}'(\text{TP}_A) = \mathcal{P}(\text{TP}_A) \pm \epsilon$. In case $a \in (A, 1)$, and $b \notin (A, 1)$, then $\mathcal{P}'(A, 1) = \mathcal{P}(A, 1) - \epsilon$.
649 We know that $\mathcal{P}'(A, 1) = \mathcal{P}'(\text{TP}_A) + \mathcal{P}'(\text{FN}_A)$. Either $a$ lies completely in $\text{TP}_A$, completely in
650 $\text{FN}_A$, or in both (if we are randomizing over the cell containing $a$). We first consider the first case,
651 where $\mathcal{P}'(\text{TP}_A) = \mathcal{P}(\text{TP}_A) - \epsilon$.

$$
\begin{aligned}
|\text{TPR}_A(f, \mathcal{P}) - \text{TPR}_A(f, \mathcal{P}')| &= \left| \frac{\mathcal{P}(\text{TP}_A)}{\mathcal{P}(A, 1)} - \frac{\mathcal{P}(\text{TP}_A) - \epsilon}{\mathcal{P}(A, 1) - \epsilon} \right| \\
&= \left| \frac{\mathcal{P}(\text{TP}_A)\mathcal{P}(A, 1) - \mathcal{P}(\text{TP}_A)\epsilon - \mathcal{P}(\text{TP}_A)\mathcal{P}(A, 1) + \mathcal{P}(A, 1)\epsilon}{\mathcal{P}(A, 1)\left(\mathcal{P}(A, 1) - \epsilon\right)} \right| \\
&= \left| \frac{\mathcal{P}(A, 1)\epsilon - \mathcal{P}(\text{TP}_A)\epsilon}{\mathcal{P}(A, 1)\left(\mathcal{P}(A, 1) - \epsilon\right)} \right| \\
&= \epsilon \left| \frac{\mathcal{P}(\text{FN}_A)}{\mathcal{P}(A, 1)\left(\mathcal{P}(A, 1) - \epsilon\right)} \right| \\
&\leq \epsilon \left| \frac{1}{\mathcal{P}(A, 1) - \epsilon} \right| \\
&= \epsilon \left| \frac{1}{\mathcal{P}'(A, 1)} \right| \\
&\leq \epsilon \left( \frac{1}{\min(\mathcal{P}(A, 1), \mathcal{P}'(A, 1))} \right)
\end{aligned}
$$

652 We now consider the second case, where $\mathcal{P}'(\text{FN}_A) = \mathcal{P}(\text{FN}_A) - \epsilon$.

$$
\begin{aligned}
|\text{TPR}_A(f, \mathcal{P}) - \text{TPR}_A(f, \mathcal{P}')| &= \left| \frac{\mathcal{P}(\text{TP}_A)}{\mathcal{P}(A)} - \frac{\mathcal{P}'(\text{TP}_A)}{\mathcal{P}'(A, 1)} \right| \\
&= \left| \frac{\mathcal{P}(\text{TP}_A)}{\mathcal{P}(A, 1)} - \frac{\mathcal{P}(\text{TP}_A)}{\mathcal{P}(A, 1) - \epsilon} \right| \\
&= \left| \frac{\mathcal{P}(\text{TP}_A)\mathcal{P}(A, 1) - \mathcal{P}(\text{TP}_A)\epsilon - \mathcal{P}(\text{TP}_A)\mathcal{P}(A, 1)}{\mathcal{P}(A, 1)\left(\mathcal{P}(A, 1) - \epsilon\right)} \right| \\
&= \left| \frac{\mathcal{P}(\text{TP}_A)\epsilon}{\mathcal{P}(A, 1)\left(\mathcal{P}(A, 1) - \epsilon\right)} \right| \\
&\leq \epsilon \left| \frac{1}{\mathcal{P}(A, 1) - \epsilon} \right| \\
&= \epsilon \left| \frac{1}{\mathcal{P}'(A, 1)} \right| \\
&\leq \epsilon \left( \frac{1}{\min(\mathcal{P}(A, 1), \mathcal{P}'(A, 1))} \right) \quad (9)
\end{aligned}
$$

653 It is easy to see that in the third case, where $a$ lies in both $\text{TP}_A$ and $\text{FN}_A$, $\Delta\text{TPR}_A$ is bounded by the
654 max value of $\Delta\text{TPR}_A$ of cases 1 and 2.

655 Here we argued for when $(A, 1)$ loses mass. We can similarly argue the case where $(A, 1)$ gains
656 mass, giving us an identical bound. Also, here we argued for group $A$, and an identical argument for
657 $D$ shows that

$$
|\text{TPR}_A(f, \mathcal{P}) - \text{TPR}_A(f, \mathcal{P}')| \leq \epsilon \left( \frac{1}{\min(\mathcal{P}(D, 1), \mathcal{P}'(D, 1))} \right) \quad (10)
$$

658 Plugging Equations 9 and 10 into Equation 8, we get that

$$
|\text{Unf}_{\text{EO}}(f, \mathcal{P}) - \text{Unf}_{\text{EO}}(f, \mathcal{P}')| \leq \epsilon \left( \frac{1}{\min(\mathcal{P}(D, 1), \mathcal{P}'(D, 1))} \right) + \epsilon \left( \frac{1}{\min(\mathcal{P}(D, 1), \mathcal{P}'(D, 1))} \right)
$$

659 $\qquad\qquad\qquad\qquad\qquad\qquad\qquad\qquad\qquad\qquad\qquad\qquad\qquad\qquad\qquad\qquad\qquad\qquad\qquad\square$

660 Now we prove our main result (Theorem 2).

661 *Proof of Theorem 2.* Following the proof of Theorem 1, by Lemma 1 and transitivity, it suffices to
662 show the theorem statement for the case where the transition from $\mathcal{P}$ to $\mathcal{P}'$ is elementary in that the
663 only difference between the two distributions is that there are two elements $a$ and $b$ that have $\epsilon$ more
664 mass and $\epsilon$ less mass, respectively, in $\mathcal{P}$ as compared to $\mathcal{P}'$ (all other elements have the same mass
665 in the two distributions). So, in the remainder of the proof, we only consider elementary transitions.

666 Consider the transfer of $\epsilon$ mass from $a$ to $b$ in a continuous manner. During this process, either the
667 cell corresponding to element $a$ will monotonically increase in score or monotonically decrease in
668 score[8]. The same holds for the cell corresponding to element $b$. The scores of all other cells will
669 remain the same. In the following argument, we assume that the score of the cell of $a$ decreases
670 monotonically and that of $b$ increases monotonically. All of the arguments are analogous for the
671 remaining three cases.

672 Let $f$ denote the EO-fair BOC for the current distribution $\mathcal{P}$ at any instant in this mass transfer
673 process ending in distribution $\mathcal{P}'$. As the mass transfer proceeds, we analyze how the EO-fair BOC
674 changes from $f_{\mathcal{P}}^{\text{EO}}$ to $f_{\mathcal{P}'}^{\text{EO}}$. We consider the largest mass transfer $\delta\epsilon$ until one of the two following
675 events occur.

676      1. Equal-score event: The cell of $a$ has the same score as the adjacent cell lower in the sorted
677         order or the cell of $b$ has the same score as the adjacent cell higher in the sorted order.

678      2. Threshold event: The score of a merged cell containing $a$ or $b$ becomes exactly 0.5.

679 Note that by the choice of $\delta\epsilon$, during the transfer $\delta\epsilon$, all the cells remain in the same order in both
680 groups; furthermore, all masses and scores of all cells other than the ones containing $a$ or $b$ remain
681 the same during the transfer. By Claim 3,

$$\delta\text{Unf}_{\text{EO}} = \left| \text{Unf}_{\text{EO}}(f_{\mathcal{P}}^{\text{EO}}, \mathcal{P}) - \text{Unf}_{\text{EO}}(f_{\mathcal{P}}^{\text{EO}}, \mathcal{P}') \right|$$
$$\leq \delta\epsilon \left( \frac{1}{\mathcal{P}(A,1)} + \frac{1}{\mathcal{P}(D,1)} \right).$$

682 Since $\text{Unf}_{\text{EO}}(f_{\mathcal{P}}^{\text{EO}}(\mathcal{P})) = 0$, we know that $\delta\text{Unf}_{\text{EO}} = \text{Unf}_{\text{EO}}(f_{\mathcal{P}}^{\text{EO}}, \mathcal{P}') =$
683 $\left| \text{TPR}_A(f_{\mathcal{P}}^{\text{EO}}, \mathcal{P}') - \text{TPR}_D(f_{\mathcal{P}}^{\text{EO}}, \mathcal{P}') \right|$. Consider the cell $q$ that is split by the threshold corre-
684 sponding to $f$ (for now, assume $q \in D$). Since neither the equal-score event nor the 0.5-score
685 event occur, we see that after the transition, $f_{\mathcal{P}}^{\text{EO}}$ has $\delta\text{Unf}_{\text{EO}}$ difference in TPR between groups.
686 To modify $f_{\mathcal{P}}^{\text{EO}} \to f_{\mathcal{P}'}^{\text{EO}}$, we therefore need to move to move the boundary at $q$ so that TPR in both
687 groups align and EO is satisfied (the classifier $f$ remains the same apart from its action on $q$).
688 The change in function ($|\Delta f(q)|$) of element $q$ is bounded by $\delta\text{Unf}_{\text{EO}} \frac{\mathcal{P}(D,1)}{\mathcal{S}(q)\mathcal{P}(q)}$, after scaling (since
689 $\mathcal{P}(D,1)\delta\text{Unf}_{\text{EO}} = |\Delta f(q)| \mathcal{P}(q)\mathcal{S}(q)$). If $f'$ denotes the EO-Fair BOC for the distribution at the
690 end of the $\delta\epsilon$ mass transfer (just prior to any of the two events), then by Lemma 2, the change in
691 accuracy of the optimal fair classifier is bounded by

$$|\text{Acc}(f, \mathcal{P}) - \text{Acc}(f', \mathcal{P}')| \leq |\mathcal{P}(q)(2\mathcal{S}(q) - 1)\Delta f(q)| + \delta\epsilon$$
$$\leq \delta\epsilon \left( \frac{1}{\min(\mathcal{P}(A,1), \mathcal{P}'(A,1))} + \frac{1}{\min(\mathcal{P}(D,1), \mathcal{P}'(D,1))} \right) \frac{\mathcal{P}(D,1) \, |(2\mathcal{S}(q) - 1)|}{\mathcal{S}(q)}$$
$$\tag{11}$$
$$\leq \delta\epsilon \left( \frac{1}{\min(\mathcal{P}(A,1), \mathcal{P}'(A,1))} + \frac{1}{\min(\mathcal{P}(D,1), \mathcal{P}'(D,1))} \right) \frac{\mathcal{P}(D)}{\mathcal{S}(q)} + \delta\epsilon,$$
$$\tag{12}$$

692 where the last equation follows by monotonicity. Note that $\frac{1}{\mathcal{S}(q)}$ can potentially blow up, and we
693 would like to bound it. Notice that since by assumption, $f$ splits $q$ in the middle, we know that there
694 is a portion of $q$ that is accepted. Hence, the weighted score of a merged cell involving $q$ (say $\mathcal{C}_q$) has

---

[8]In case the cell corresponding to $a$ has score of 0 or 1, it's score will remain unchanged, and this case is
trivially covered by our argument.

score above the threshold of $0.5$. Let $C_q$ contain some element $t$ from group $A$, and denote length of group $z$ in $C_q$ by $l_z$. Since the TPR of both components are equal, we know that

$$\frac{\mathcal{P}(A)l_A\mathcal{S}(t)}{\mathcal{P}(A,1)} = \frac{\mathcal{P}(D)l_D\mathcal{S}(q)}{\mathcal{P}(D,1)} \tag{13}$$

Also, since $\mathcal{S}(C_q) \geq 0.5$, we know that

$$\mathcal{P}(A)l_A\mathcal{S}(t) + \mathcal{P}(D)l_D\mathcal{S}(q) \geq \frac{\mathcal{P}(A)l_A + \mathcal{P}(D)l_D}{2} \tag{14}$$

Combining Equations 13, and 14, and after a bunch of simplification, we get that

$$\frac{1}{\mathcal{S}(q)} \leq \frac{2\mathcal{P}(1)}{\mathcal{P}(D,1)} - \frac{\mathcal{P}(A,1)}{\mathcal{P}(D,1)\mathcal{S}(t)} \tag{15}$$

$$\leq \frac{2\mathcal{P}(1)}{\mathcal{P}(D,1)} \tag{16}$$

Where the second equation follows because $\mathcal{S}(t) \geq 0$. Plugging Equation 16 into Equation 12, we get that

$$|\text{Acc}(f,\mathcal{P}) - \text{Acc}(f',\mathcal{P}')| \leq \delta\epsilon \left( \frac{1}{\min(\mathcal{P}(A,1),\mathcal{P}'(A,1))} + \frac{1}{\min(\mathcal{P}(D,1),\mathcal{P}'(D,1))} \right) 2\mathcal{P}(1) + \delta\epsilon$$

$$\leq \delta\epsilon \left( \frac{1}{\min(\mathcal{P}(A,1),\mathcal{P}'(A,1))} + \frac{1}{\min(\mathcal{P}(D,1),\mathcal{P}'(D,1))} \right) 2\max(\mathcal{P}(1),\mathcal{P}'(1)) + \delta\epsilon$$
$$\text{(monotonicity)}$$

The handling of the equal-score and threshold events is identical to that in the proof of Theorem 1. We repeat here for convenience.

1. Equal-score event: If the cell of $a$ has the same score as the adjacent cell lower in the sorted order, then we swap the two cells so that the cell of $a$ is lower in the order. Similarly, if the cell of $b$ has the same score as the adjacent cell higher in the order, then we swap the two cells so that the cell of $b$ is higher in the order. We update the classifier $f$ and note that this change has no impact on the accuracy of $f$.

2. Threshold event: The score of a merged cell containing $a$ or $b$ becomes exactly $0.5$. We include the merged cell in the classifier $f$, again without changing accuracy.

Thus, between any two occurrences of these events, the change in accuracy is bounded by an amount proportional to the mass transfer; when we reach these occurrences, the mass transfer is paused, the BOC changes without any change in accuracy. Furthermore, at every occurrence of the event, one of these three events happen: the cell containing $a$ moves down in the order, the cell containing $b$ moves up in the order, or an additional merged cell is placed above the threshold. Since the number of times these events can occur is upper bounded by the number of cells in the two groups, this process is finite. Therefore, adding over all the $\delta\epsilon$ mass transfers, we obtain the desired bound on the change in accuracy between the BOC's for $\mathcal{P}$ and $\mathcal{P}'$, thus completing the proof of the theorem. $\square$

## C.3 Predictive Equality

We can obtain analogous results for Predictive Equality from the same proof techniques as that of Equal Opportunity (since we can just reverse the roles of the labels 0 and 1 in EO to get results for PE). Hence, we only discuss the proofs for EO, and state the analogous results for PE below without proof.

**Claim 7** (PE Shift for a Fixed Hypothesis). *Given distributions $\mathcal{P}, \mathcal{P}'$, such that $TV(\mathcal{P}, \mathcal{P}') \leq \epsilon$, and any hypothesis $f$, it holds that*

$$|Unf_{PE}(f,\mathcal{P}) - Unf_{PE}(f,\mathcal{P}')| \leq \epsilon \left( \frac{1}{\min(\mathcal{P}(A,0),\mathcal{P}'(A,0))} + \frac{1}{\min(\mathcal{P}(D,0),\mathcal{P}'(D,0))} \right).$$

**Theorem 3** (Robustness of PE-Fair BOC)**.** *Given distributions $\mathcal{P}, \mathcal{P}'$, such that $TV(\mathcal{P}, \mathcal{P}') = \epsilon$, we have that*

$$\left| Acc(f_{\mathcal{P}}^{PE}, \mathcal{P}) - Acc(f_{\mathcal{P}'}^{PE}, \mathcal{P}') \right| \le \epsilon \left( 1 + 2 \max(\mathcal{P}(0), \mathcal{P}'(0)) \left( \frac{1}{\min(\mathcal{P}(A,0), \mathcal{P}'(A,0))} + \frac{1}{\min(\mathcal{P}(D,0), \mathcal{P}'(D,0))} \right) \right),$$

*where $f_{\mathcal{P}}^{PE}, f_{\mathcal{P}'}^{PE}$ are the PE-Fair BOC's on $\mathcal{P}, \mathcal{P}'$ respectively.*

**Corollary 3.** *Given distributions $\mathcal{P}, \mathcal{P}'$, such that $TV(\mathcal{P}, \mathcal{P}') = \epsilon$, we have that*

$$\left| Acc(f_{\mathcal{P}}^{PE}, \mathcal{P}) - Acc(f_{\mathcal{P}'}^{PE}, \mathcal{P}) \right| \le 2\epsilon \left( 1 + \max(\mathcal{P}(0), \mathcal{P}'(0)) \left( \frac{1}{\min(\mathcal{P}(A,0), \mathcal{P}'(A,0))} + \frac{1}{\min(\mathcal{P}(D,0), \mathcal{P}'(D,0))} \right) \right),$$

*where $f_{\mathcal{P}}^{EO}, f_{\mathcal{P}'}^{EO}$ are the EO-Fair BOC's on $\mathcal{P}, \mathcal{P}'$ respectively*

