# OpenReview forum: "Optimal Fair Learning Robust to Adversarial Distribution Shift"
_NeurIPS.cc/2025/Workshop/Reliable_ML — NeurIPS 2025 - Reliable ML Workshop_

### Official Review · Reviewer_Ck6B · 2025-09-13
**A theoretical paper**

**Rating:** 6
**Confidence:** 1

**Review:**

**Summary.**
This paper shows that the deterministic Fair BOC is not robust to adversarial noise, and proves the robustness of randomized Fair BOC’s. They also show the  multiple advantages brought by randomization.

**Strengths.**
This paper provides sufficient theoretical evidence to support its claims.

**Weaknesses / Limitations.**
The empirical evidence is not presented in the paper.

**Suggestions.**
The presentation of formulas could be improved so that they won’t violate the page margins.

The writing of Section 1 could be improved. The current version is more like the related work section.

**Ethics.**
None.

---

### Official Review · Reviewer_UK1c · 2025-09-19
**Review of #185**

**Rating:** 7
**Confidence:** 2

**Review:**

Summary:
The paper studies robustness of fair Bayes-optimal classification. It shows (i) a simple counterexample where the deterministic Demographic Parity (DP)-fair Bayes-optimal classifier (BOC) is not robust—tiny TV perturbations can force accuracy to differ by ~0.5 (Claim 1), and (ii) a positive result: the randomized fair BOC satisfies a local Lipschitz robustness guarantee for DP (Theorem 1), with extensions to Equal Opportunity (EO) and Predictive Equality (PE). A key structural insight is that the optimal randomized fair rule is a mass-threshold classifier that needs to randomize on at most one cell (i.e., it is “nearly deterministic”). The paper further shows that randomization can strictly improve accuracy over deterministic fair rules and that computing a deterministic fair BOC is NP-complete, whereas the randomized fair BOC admits a simple, efficient construction on discrete domains.


Strengths:
The paper gives a clean counterexample which exhibits how a tiny TV perturbation can destroy robustness for deterministic DP-fair classifiers. The constructive, geometric characterization of the randomized fair BOC is elegant and practically appealing. Finally, the narrative ties nicely to prior robustness work (Kearns–Li) thus situating the contribution well.

Weaknesses:
The Lipschitz constants in the robustness bounds blow up when a group’s mass is small (the paper notes this), so the guarantees can degrade precisely in high-imbalance regimes where fairness concerns are acute; the paper could benefit from some discussion of mitigation or alternative calibrations. The scope is primarily discrete domains with exact fairness and TV as the shift metric, with no empirical illustrations; given the motivating applications (e.g., toxicity/hate-speech annotation biases), even small synthetic plots demonstrating the “one-cell” mechanism and the empirical tightness of constants would strengthen the paper.